# Metastable spiking networks in the replica-mean-field limit

**Luyan Yu**[1], **Thibaud O. Taillefumier**[2,3]*

**1** Department of Physics, University of Texas at Austin, Austin, Texas, United States of America,
**2** Department of Mathematics, University of Texas at Austin, Austin, Texas, United States of America,
**3** Department of Neuroscience, University of Texas at Austin, Austin, Texas, United States of America

* ttaillef@austin.utexas.edu

## Abstract

Characterizing metastable neural dynamics in finite-size spiking networks remains a daunting challenge. We propose to address this challenge in the recently introduced replica-mean-field (RMF) limit. In this limit, networks are made of infinitely many replicas of the finite network of interest, but with randomized interactions across replicas. Such randomization renders certain excitatory networks fully tractable at the cost of neglecting activity correlations, but with explicit dependence on the finite size of the neural constituents. However, metastable dynamics typically unfold in networks with mixed inhibition and excitation. Here, we extend the RMF computational framework to point-process-based neural network models with exponential stochastic intensities, allowing for mixed excitation and inhibition. Within this setting, we show that metastable finite-size networks admit multistable RMF limits, which are fully characterized by stationary firing rates. Technically, these stationary rates are determined as the solutions of a set of delayed differential equations under certain regularity conditions that any physical solutions shall satisfy. We solve this original problem by combining the resolvent formalism and singular-perturbation theory. Importantly, we find that these rates specify probabilistic pseudo-equilibria which accurately capture the neural variability observed in the original finite-size network. We also discuss the emergence of metastability as a stochastic bifurcation, which can be interpreted as a static phase transition in the RMF limits. In turn, we expect to leverage the static picture of RMF limits to infer purely dynamical features of metastable finite-size networks, such as the transition rates between pseudo-equilibria.

## Author summary

Electrophysiological recordings show that neural circuits process information by dynamically switching between quasi-stationary states, whereby neurons exhibit sustained, stereotypic activity. Such alternations of stable and unstable bouts of activity is referred to as neural metastability. The observation of metastability supports the view that neural computations are implemented by sequences of input-dependent transitions between quasi-stationary states. Therefore, understanding neural computation conceptually hinges on

**Data Availability Statement:** The source code and data used to produce the results and analyses presented in this manuscript are available at GitHub: https://github.com/yuluyan/rmf-metastability.

**Funding:** L.Y. was supported by the Provost's graduate excellence fellowship (https://ph.utexas.

edu/graduate/prospective-graduate-students/
financial-support) and the Center of Theoretical and
Computational Neuroscience (https://ctcn.utexas.
edu/) at the University of Texas at Austin. T.T. was
supported by Alfred P. Sloan Research Fellowship
(https://sloan.org/fellowships/) grant no. FG-2017-
9554 and CRCNS award (https://beta.nsf.gov/
funding/opportunities/collaborative-research-
computational-neuroscience-crcns) grant no.
DMS-2113213 from National Science Foundation.
The funders had no role in study design, data
collection and analysis, decision to publish, or
preparation of the manuscript.

**Competing interests:** The authors have declared
that no competing interests exist.

characterizing metastable dynamics in biophysically relevant network models. Modeling-wise, metastable dynamics can only emerge in finite-size neural networks, for which irreducible neural variability controls the rate of transition between various quasi-stationary states. Unfortunately, the quantitative analysis of neural networks typically requires simplifying assumptions that effectively erase finite-size induced metastability. To remedy this point, we apply a "multiply-and-conquer" approach that consider neural networks made of infinitely many copies of the original network of interest. This setting allows us to introduce simplifying assumptions that render the dynamics of certain biophysically relevant networks tractable, while remaining predictive of metastability in the finite-size original network.

## Introduction

One of the striking features of neural activity is its high degree of trial-to-trial variability in response to identical stimuli [1, 2]. In all generality, there are two nonexclusive explanations for such variability: neural variability can reflect fluctuations of unmonitored variables, such as global attention state or cross-sensory influences [3, 4]; or neural variability can arise from inherently noisy transduction mechanisms, such as photon counting noise or faulty synaptic transmissions [5, 6]. Independent of its origin, neural variability propagates throughout neural circuits in the form of seemingly stochastic spiking activity. In turn, these neural circuits must shape part of this variability [7]. Owing to the high degree of connectivity observed in cortical circuits, a prevailing hypothesis is that a neuron's response is primarily shaped by its mean input drive, computed as an average over many equally contributing synapses [8]. However, contrary to this view, some experimental evidence supports that synaptic inputs can impact the neural response individually rather than via population averages. For instance, calcium imaging of dendritic trees in the visual cortex has revealed that synapses impinging on the same neuron are tuned to largely distinct features of the inputs [9, 10]. This suggests that only a fraction of the otherwise large number of synaptic inputs is active when processing visual information via subthreshold integration. Moreover, like many other components of neural activity, synaptic currents have been shown to be approximately lognormally distributed in cortex [11, 12]. Such heavy-tailed distributions also suggest that a small number of large synapses primarily shapes subthreshold integration. Finally, the post-spiking reset mechanisms taking place in each neuron formally implement a feedback of the neuron's spiking activity onto itself [13]. In this view, albeit not a synaptic input, a neuron's own spiking represents the single input that most reliably impacts subthreshold integration. These observations support that a significant part of the neural variability arises from the discrete processing of a limited number of stochastic spiking inputs. As a result of this discrete processing, understanding quantitatively variability remains a conundrum in realistic neural networks. This limitation is especially concerning as variability has been recognized as an integral part of neural computations rather than a mere nuisance to faithful information processing [14–16]. For instance, neural variability is a key determinant of the metastable dynamics thought to support computations in neural networks [17, 18].

A primary hurdle to understanding neural variability is perhaps the lack of mechanistic network models for which variability can be quantified in relation to a few biophysically relevant neural features. Quantifying variability in stochastic network models often relies on drastic simplifying approximations, typically obtained in the thermodynamic-mean-field (TMF) limit [19–22]. This limit considers infinite-size networks whereby neurons receive a large number

of vanishingly small synaptic inputs, with synaptic strengths typically scaling in inverse proportion of the number of inputs. By virtue of their definition, TMF limits fail to capture the variability of finite-size neural networks when stochasticity is shaped by the discrete nature of the interacting neuronal components. This is because TMF limits substitute a deterministic mean-field drive for fundamentally stochastic neural inputs, thereby erasing some of the variability of the original, finite-size network. To better capture finite-size effects in neural variability, it would be useful to have mean-field models that preserve the stochastic nature of the neural inputs. In [23], we introduce such models within the replica-mean-field (RMF) framework. This framework elaborates on a *multiply-and-conquer* approach drawing on ideas from the theory of communication systems [24, 25] rather than from statistical physics [26, 27]. The focus of the RMF framework is the analysis of a limit network, the so-called RMF network, whose dynamics approximates the activity of the original, finite network of interest. This RMF limit is obtained by making infinitely many replicas of the original network and by implementing a randomized routing of the interactions across replicas. Randomizing interactions across replicas yields simplified dynamics by erasing nontrivial dependencies in RMF networks, where neurons are effectively driven by independent Poissonian bombardments [28]. Computationally, the RMF approach aims at specifying the stationary rates of these driving Poisson processes as functions of the parameters defining the original network. Such parameters naturally include, e.g., the finite neuronal population size, the finite degree of synaptic connectivity, and the finite synaptic strengths. Our introductory work [23] demonstrated that RMF limits can capture finite-size effects in the stationary dynamics of certain neural networks. However, this was only done for a restricted class of excitatory models with a limited range of dynamics, excluding metastable ones.

Here, we develop the RMF computational framework for a new class of models with mixed inhibition and excitation and we demonstrate that the RMF approach applies to an extended range of dynamics, including metastable ones. These models, referred to as the linear-exponential reset (LER) models, have three essential features: (*i*) LER neurons integrate finite-size spiking interactions via a continuously relaxing internal variable, mimicking the membrane potential; (*ii*) the instantaneous neuronal firing rate is determined as an exponential function of the neuron's internal variable; and (*iii*) the internal variable resets to base level upon spiking, thereby implementing a refractory period. Thus defined, LER models belong to a larger class of generalized linear models studied in computational neuroscience [29] and are examples of stochastic-intensity-based models. Stochastic-intensity-based models have been successful in accounting for many key problems in neural coding such as measuring the regularity of neuron spiking events [30], decoding velocity and direction from the motor cortical recordings [31] and predicting the stability of the neuronal dynamics [29]. In this work, we present analytical tools and numerical methods to calculate the stationary RMF firing rates in LER neural networks as functions of the network parameters. In the RMF setting, these rates fully parametrize the neural dynamics as sufficient statistics for the stationary distribution of the network states. Thus, we are able to derive the neural variability, i.e., the moments of the stochastic intensities and of the internal variables, from the knowledge of the firing rates alone. Such calculations can be performed in the RMF limits for any network topologies and are numerically efficient in the sense that they avoid Monte-Carlo schemes [32]. Moreover, we find that the RMF estimates agree well with the exact, event-driven simulations of the original finite-size network [33, 34].

Due to their (exponential) nonlinearity, we find that mixed networks of LER neurons are prone to metastable dynamics. Metastable networks exhibit dynamics characterized by fluctuations around pseudo-equilibrium states at small time scales and sharp transitions between pseudo-equilibria at larger time scales. Simulating finite-replica models of metastable networks

reveals that the transition rates between pseudo-equilibria typically vanish exponentially with the number of replicas. Thus, just as in TMF limits, dynamical ergodicity breaks down in infinite-size metastable RMF networks and metastability turns into multistability. Concretely, this means that the equations governing the TMF and RMF dynamics both admit multiple solutions for the stationary rates [35]. However, by contrast with TMF limits, RMF limits predict self-consistently a nontrivial stationary distribution for the inputs, and correspondingly, nonzero moments at any order. We show that the first two RMF moments can satisfactorily capture the variability of a finite, metastable network of interest. Specifically, we demonstrate this point for a bistable neural network whose structure is motivated by the study of perceptual rivalry in neuroscience [36]. Remarkably, the RMF limit can predict the variability of such networks even when the metastability originates from as few as 40 strongly interacting neurons, when finite-size effects dominate. These results are obtained when subjecting the 40 neurons to weak TMF-like excitatory inputs, modeling uncorrelated background activity [9, 37]. Elaborating on this example, we also show that our RMF approach can numerically detect the emergence of bistability as a stochastic pitchfork bifurcations [38]. We further discuss this stochastic pitchfork bifurcation, a dynamical phenomenon, as a form of static phase transitions whereby order is established across replicas. We also show numerical evidence suggesting that transition rates can indeed be inferred from quantifying neural variability in the RMF approach.

Methodology-wise, we derive the self-consistent equations for the RMF stationary rates of LER networks in the form of delay differential equations (DDEs). Unlike in standard settings, these DDEs do not come equipped with a notion of initial conditions on a delayed range to specify their solutions [39–42]. Rather, we determine these solutions solely by imposing the regularity and normalization conditions that any probabilistic model shall satisfy. To our knowledge, there are no closed-form solutions to this problem and to date, there are no standard method for numerically solving it. We develop such a method by adapting the resolvent formalism [43–45] to write the RMF stationary firing rates as divergent series. In turn, we compute the resulting rates via an iterative scheme utilizing Padé approximants summation [46, 47]. This methodology incidentally delivers two insights: (1) The RMF approach reveals that three biophysical timescales compete to shape the neuronal response, including its variability. These are the timescales associated with relaxation, with spontaneous firing, and with spiking—or rather reset. In that respect, a key insight is to recognize that a post-spiking reset acts as a randomizing feedback, which implements rate saturation while promoting independent neuronal variability. This observation is independent of modeling details, has implications in terms of network stability, and is physically interpretable within singular perturbation theory. (2) The RMF approach offers to analyze metastable systems via finite-dimensional stochastic bifurcations, which can be interpreted as static phase transitions over the replicas. This is conceptually new on two grounds as (a) even in finite-dimensional systems, stochastic bifurcations generally form infinite-dimensional processes and (b) stochasticity is erased when phase transitions are treated in classical thermodynamic limits. In this context, a natural future direction is expanding the RMF analysis to systematically predict transition rates in finite metastable network models.

The structure of the manuscript is as follows. First, we formulate the RMF framework for the network dynamics of interest and establish the corresponding RMF self-consistent equations. Second, we discuss these equations from a numerical point of view and review Padé approximants summation for numerical estimation. Third, we demonstrate our computational approach with a focus on finite-size effects in metastable dynamics and compare them with the TMF approximations. Finally, we discuss the biophysical relevance of our modeling approach and introduce some future computational extensions.

## Modeling and theory

### General approach and dynamics of interest

**Replica-mean-field framework for spiking networks.** In principle, the RMF approach can apply to any network dynamics whereby elementary network constituents—here neurons—interact via spikes. By spikes, we mean that interactions come in the form of precisely time-stamped, stereotypical, transient impulses. Fig 1a shows an example of networks evolving in response to spiking interactions. In such finite-size networks, spiking interactions generally introduce complex dependencies between neurons, with non-trivial correlation structure. Because of these complex dependencies, almost all finite-size spiking networks are analytically intractable. To circumvent this limitation, classical modeling approaches approximate the dynamics of interest via simplifying mean-field limits with trivial correlation structure, i.e., with independently firing neurons [19–22]. However, these limits are obtained by considering approximating networks of infinite size, whereby each neuron interacts with an infinite number of neurons, but via vanishingly small interactions. Thus, classical mean-field approaches erase the neural variability originating from the finite size of interactions.

The purpose of the RMF approach is to better capture neural variability by performing a modified mean-field approximation that preserves finite-size effects. Specifically, we design the RMF approach in [23, 28] so that neurons still emit spikes in response to a variable but discrete number of inputs, each with finite size, as opposed to being subjected to a deterministic average drive. In short, these RMF dynamics are obtained by assuming that each neuron receives inputs from other neurons via independent Poisson processes. This amounts to assuming that interactions occur as randomly as possible, i.e., with trivial dependencies, given that these interactions still take the form of variable, discrete, spiking events. *A priori*, it is unclear why such a simplifying assumption, commonly referred to as the Poisson hypothesis in network theory [48, 49], would lead to the well-posed dynamics of some limit physical system. Indeed, even when driven by Poissonian independent inputs, spiking neuronal models generally have non-Poissonian outputs. In other words, the Poissonian regime does not naturally stabilize in finite-size dynamics.

Fortunately, it turns out that a simple replication process allows one to build physical limit networks supporting RMF network dynamics. This replication process produces enlarged networks made of $R$ copies of the original network, each with $K$ neurons indexed by $i$, $1 \leq i \leq K$. In such $R$-replica networks illustrated in Fig 1b, when a neuron $(i, r)$ of type $i$ spikes from a replica $r$, $1 \leq r \leq R$, it interacts with neurons $(j, q_j)$, $j \neq i$, according to the same rule as in the

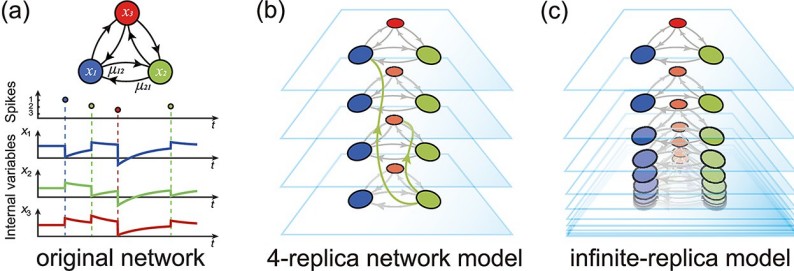

original network                4-replica network model              infinite-replica model

**Fig 1. RMF models.** Panel (a) shows the original networks of $K = 3$ neurons. Panel (b) represents the finite-replica model with $r = 4$ replicas. When a neuron spikes (e.g., green neuron), it interacts with downstream neurons sampled uniformly at random across replicas. Panel (c) shows schematically that the RMF models are obtained in the limit of an infinite number of replicas and represent infinite-size physical models supporting RMF dynamics. This figure is reproduced from [52] with permission.

original dynamics, except that each target replica $q_j$ is drawn uniformly and independently across replicas. Thus, the construction of RMF networks relies crucially on the possibility to precisely time interactions in spiking networks as it allows for the randomized routing of interactions across replicas. RMF networks are obtained by taking the limit of an infinite number of replicas $R \to \infty$ as depicted in Fig 1c. In this limit, neurons become independent as suggested by the fact that their probability to interact over a finite period of time vanishes as $1/R$. Accordingly, the law of rare (interaction) events [50] suggests that when collected across replicas, inputs to a given neuron $(i, r)$ from neurons of type $j$ are received with the same interaction size $\mu_{ij}$ according to a Poisson process. Thus, in RMF networks, neurons are allowed to individually deviate from being Poisson spike generators because randomizing interactions across replicas guarantees aggregated Poissonian spiking deliveries. These intuitive arguments can be checked numerically and shall hold for all spiking networks. In this work, we focus on rate models called intensity-based spiking networks, which comprise a large class of models for which we rigorously established the Poisson hypothesis in [51].

**Mean-field limits for a simple intensity-based spiking network.** In general, the activity of a $K$-neuron spiking network can be modeled as a $K$-dimensional stochastic point process $\{N_i(t)\}_{1 \leq i \leq K}$, where $t$ denotes time and $i$ is the neuron index [53, 54]. For each neuron $i$, the component $N_i(t)$ is specified as the counting process registering the spiking occurrences of neuron $i$ up to time $t$. In other words, $N_i(t) = \sum_k \mathbb{1}_{\{t_{i,k} \leq t\}}$, where $\{t_{i,k}\}_{k \in \mathbb{Z}}$ denotes the full sequence of spiking times of neuron $i$ and we label spikes so that $t_{i,0} \leq 0 < t_{i,1}$ for all $1 \leq i \leq K$ by convention. $\mathbb{1}_A$ denotes the indicator function of set $A$, where $\mathbb{1}_A(x) = 1$ if $x$ is in $A$ and $\mathbb{1}_A(x) = 0$ if $x$ is not in $A$. The simplest instance of such counting processes $N_i(t)$ is the Poisson process for which spike generation is governed by a deterministic, instantaneous, firing rate function $\lambda_i(t)$. Informally, the rate $\lambda_i(t)$ determines the probability of finding a spike in the infinitesimal time interval $(t, t + dt)$ as $\mathbb{P}[N_i(t + dt) > N_i(t)] = \lambda_i(t)\, dt$. A shortcoming of Poisson processes is that they do not allow for stochastic spiking inputs to individually shape the instantaneous firing rate, as should be the case in realistic stochastic spiking models. To address this point, the instantaneous firing rate $\lambda_i(t)$ must be modeled as a stochastic process, whose probability law depends on the past states of the network. Formally, this corresponds to defining $\lambda_i(t)$ as the stochastic intensity of the point process $N_i(t)$ [55, 56]. Then, specifying the dynamics of the corresponding intensity-based networks consists in mechanistically relating the joint law of $\{\lambda_i(t)\}_{1 \leq i \leq K}$ to the past spiking history of the network $\{N_i(s)\}_{1 \leq i \leq K, s \leq t}$, possibly given some initial condition $\{\lambda_i(0)\}_{1 \leq i \leq K}$.

In the absence of a reset mechanism, the simplest mechanistic intensity-based models are perhaps the celebrated self-excited linear Hawkes processes [57, 58]. These processes are specified by the linear integral equations

$$\lambda_i(t) = h_i + \sum_{j \neq i} \mu_{ij} \int_{-\infty}^{t} e^{-(t-s)/\tau_i} N_j(ds)\,, \tag{1}$$

where we only consider excitatory interactions: $\mu_{ij} \geq 0$. As they only involve exponential integrands with neuron-specific time constants $\tau_i$, the above integral equations are equivalent to the perhaps more familiar stochastic differential equation

$$\lambda_i(t) = \lambda_i(0) + \underbrace{\frac{1}{\tau_i} \int_0^t (h_i - \lambda_i(s))\, ds}_{\text{relaxation}} + \overbrace{\sum_{j \neq i} \mu_{ij} N_j(t)}^{\text{interaction}}\,, \tag{2}$$

which clearly separates the relaxation term and the interaction terms. According to this

equation, in the absence of interactions, the rate $\lambda_i(t)$ relaxes to its base value $h_i$ with time constant $\tau_i$, whereas $\lambda_i(t)$ instantaneously increases by an amount $\mu_{ij}$ whenever neuron $j \neq i$ spikes. Given certain stability conditions precluding rate explosion, it is known that linear Hawkes models admit stationary dynamics. For such dynamics, the stationary rates $\beta_i = \mathbb{E}[\lambda_i]$ satisfy a system of linear equations

$$\beta_i = h_i + \tau_i \sum_{j \neq i} \mu_{ij} \beta_j \,. \tag{3}$$

This system can be derived by taking the stationary expectation of (1), using the fact that we have $\mathbb{E}[N_i(t)] = \beta_i t$ at stationarity. Then, the stability condition amounts to imposing that the above system admits a set of nonnegative rates $\{\beta_i\}_{1 \leq i \leq K}$ as solution.

Albeit linear, Hawkes stationary processes have nontrivial correlation structures due to finite-size interactions. In view of this, let us use Hawkes processes as a concrete example to compare approximations via the TMF limit and the RMF limit. In the TMF limit, each neuron is subjected to a deterministic drive obtained by averaging over an infinite-size network via the law of large numbers. Concretely, this means that one should substitute $\beta_j t$ for the original Hawkes processes $N_j(t)$ in (1). This implies that the rates $\lambda_i$ are also deterministic, and consistently setting their constant value to be $\beta_i$ directly yields the exact system of equations (3). However, all variability of the rates $\lambda_i$ is erased in the process. By contrast, the RMF limit provides us with a Poissonian approximation obtained by substituting independent Poisson processes $P_j(t)$ with rate $\beta_j$ for the original Hawkes processes $N_j(t)$ in (1). Concretely, this means that

$$\lambda_i = h_i + \sum_{j \neq i} \mu_{ij} \int_{-\infty}^{0} e^{s/\tau_i} P_j(\mathrm{d}s) \,, \tag{4}$$

where by stationarity, there is no lack of generality to choose $\lambda_i = \lambda_i(0)$. Thus, by contrast with the TMF limit, the RMF limit provides us with a stochastic approximation for the rates. Taking the stationary expectation of (4) also yields the exact system of equations (3) for the mean firing rates, whereas the rate variability can be estimated as

$$\mathbb{V}[\lambda_i] = \sum_{j \neq i} \mu_{ij}^2 \int_{-\infty}^{0} e^{2s/\tau_i} \mathbb{E}\left[P_j(\mathrm{d}s)\right] = \frac{\tau_i}{2} \sum_{j \neq i} \mu_{ij}^2 \beta_j \,, \tag{5}$$

by virtue of Campbell's formula for Poisson processes [55, 56]. The above simple treatment illustrates the distinction between TMF and RMF limits for linear Hawkes dynamics. Both TMF and RMF limits recover the exact system of equations (3). This is a special feature of linear Hawkes dynamics, and for more general intensity-based network dynamics, we shall expect TMF or RMF limits to only yield approximate mean-field relations. However, only the RMF limit provides us with consistent variability estimates, obtained at the cost of imposing a trivial correlation structure via the Poisson hypothesis. Here our goal is to quantify neural variability in the RMF limits of certain class of nonlinear intensity-based spiking networks, referred to as linear-exponential reset (LER) networks.

**Finite-size linear-exponential reset neuronal dynamics.** The LER neuron model is based on the class of linear-nonlinear Poisson (LNP) neurons [59–61], with the addition of individual post-spiking reset rules [62, 63]. Such models of LNP neurons with reset were formally introduced as the Galves-Löcherbach models [64]. In the LER network, the stochastic intensity $\lambda_i(t)$ is exponentially related to a neuron-specific internal variable $x_i(t)$, which integrates past

neuronal interactions. Specifically, we have

$$\lambda_i(t) = h_i e^{a_i x_i(t)} , \tag{6}$$

where $h_i$ and $a_i$ are positive constants. The dynamics of the internal variable $x_i(t)$ is as follows. Whenever neuron $i$ spikes, $x_i(t)$ instantaneously resets to zero, erasing all memory effects at the individual neuron's level. At the same time, neurons $j \neq i$ register the instantaneous deliveries of Dirac-delta impulses in their internal variables $x_j(t)$ via synaptic strengths $\mu_{ij}$. This registration proceeds in a leaky fashion so that $x_i(t)$ obeys the linear stochastic differential equation

$$x_i(t) = x_i(0) - \underbrace{\frac{1}{\tau_i} \int_0^t x_i(s)\, \mathrm{d}s}_{\text{relaxation}} + \overbrace{\sum_{j \neq i} \mu_{ij} N_j(t)}^{\text{interaction}} - \underbrace{\int_0^t x_i(s^-)\, N_i(\mathrm{d}s)}_{\text{reset}} , \tag{7}$$

where $\tau_i$ denotes the relaxation time of the neuronal internal variable to zero. Observe that the last integral term in Eq (7) implements the post-spiking reset rule to zero. Moreover, note that in view of Eq (7), $h_i$ appears as the (nonzero) base spiking rate of neuron $i$, whereas $a_i$ models its excitability.

Given (6), Eq (7) fully specifies the dynamics of LER networks in the same fashion as (2) does for linear Hawkes processes. It is also instructive to specify LER dynamics via a stochastic integral equation akin to (1), which directly relates the stochastic intensities $\{\lambda_i(t)\}_{1 \leq i \leq K}$ to the past spiking history of the network $\{N_i(s)\}_{1 \leq i \leq K, s \leq t}$. Such an integral equation reads

$$\lambda_i(t) = h_i \exp\left( a_i \sum_{j \neq i} \mu_{ij} \int_{(l_i(t), t]} e^{-(t-s)/\tau_i} N_j(\mathrm{d}s) \right) , \tag{8}$$

where $l_i(t)$ is the last time neuron $i$ spikes before $t$: $l_i(t) = \sup_k\{t_{i,k} \geq 0 | t_{i,k} \leq t\}$. A merit of the above equation is to exhibit the finite memory of individual neuronal dynamics with reset. To see this, observe that $\lambda_i(t)$ only depends on the past spiking history of the network starting from $l_i(t)$, i.e., on $\{N_i(s)\}_{1 \leq i \leq K, l_i(t) < s \leq t}$. Another merit of Eq (8) is to reveal that the reset mechanism implements a stochastic feedback that effectively precludes rate explosions. This follows from the fact that the last spiking time $l_i(t)$ is a stochastic variable, even in mean-field limits, and from the fact that the larger the rate, the shorter the memory length over which spiking inputs are integrated. This is because the memory length is bounded by the interspike intervals $t_{i,k+1} - t_{t,k}$, which on average, scales in inverse proportion to the mean firing rate: $\beta_i \mathbb{E}[t_{i,k+1} - t_{t,k}] = 1$. One final merit of Eq (8) is that it can generalize to spiking interactions of any shapes and to arbitrary rate dependency. However, in view of obtaining tractable RMF limits, we restrict ourselves to exponential nonlinearities, for which the specification via stochastic differential Eq (7) is possible.

To sum up in more concrete terms, we model a neural network as a directed weighted graph whose nodes are neurons and whose directed edges are synaptic weights $\mu_{ij}$ from neuron $j$ to $i$. Within the network, the state of a neuron $i$ is given by its internal variable $x_i$, which determines the neuronal instantaneous firing rate $\lambda_i$ via the exponential relation Eq (6). The individual neuronal dynamics consists of three components: (*i*) interaction, (*ii*) reset, and (*iii*) relaxation. (*i*) When neuron $i$ fires, $x_j, j \neq i$, updates to $x_j + \mu_{ji}$. (*ii*) At the same time, $x_i$ resets to zero. (*iii*) In between spikes, each $x_i(t)$ relaxes toward zero with time constant $\tau_i$. Thus specified, the dynamics of LER networks defines a continuous-time Markov chain. Considerations from the regenerative theory of Markov chains show that in the presence of relaxation, i.e., whenever $\max_i \tau_i < \infty$, LER network dynamics are ergodic [23, 65]. This means that collecting

the typical states of the networks defines a unique stationary distribution, independent of initial conditions.

## Development of the computational framework

**Replica mean-field limit for networks of exponential neurons.** A benefit of considering LER models is that they naturally accommodate inhibition by allowing for negative synaptic weight $\mu_{ij} < 0$. This is by contrast with linear models considered in [23] for which including inhibition conflicts with the required nonnegativity of the rate functions $\lambda_i(t)$. Unfortunately, just as for their linear counterparts, an exact analytical treatment of the finite-size LER models hinders on the complex structure of the activity correlations. This limitation motivates considering LER networks in the RMF limit, which comprises an infinite number of replicas of the original systems [52]. By original system, we mean the $K$-neuron network with LER dynamics described by Eq (7). The $R$-replica model is obtained by considering that a neuron $i$ within each replica $r$, $1 \leq r \leq R$, follows the same autonomous dynamics as that of neuron $i$ in the original system. However, the difference with the original system is that upon spiking, a neuron $i$ from replica $r$ interacts with neurons $(j, q)$, $j \neq i$, via the original weights $\mu_{ji}$ but in replicas $q$, chosen uniformly at random. Formally, this corresponds to the following set of stochastic equations

$$
\begin{aligned}
x_{i,r}(t) = x_{i,r}(0) - \frac{1}{\tau_i} \int_0^t x_{i,r}(s)\, \mathrm{d}s \\
+ \sum_q \sum_{j \neq i} \mu_{ij} \int_0^t \mathbb{1}_{\{v_{q,ij}(s)=r\}}\, N_{j,q}(\mathrm{d}s) - \int_0^t x_{i,r}(s^-)\, N_{i,r}(\mathrm{d}s)\,,
\end{aligned}
\tag{9}
$$

where for all $s \geq 0$, $1 \leq r, q \leq R$, $1 \leq i \neq j \leq K$, $v_{r,ij}(s)$ are independent random variables uniformly distributed over $\{1, \ldots, R\}$. These random variables are routing addresses specifying that when neuron $(j, r)$ spikes at time $s$, it targets a neuron of type $i$ in replica $v_{r,ij}(s)$.

Intuitively, the randomization of interactions present in Eq (9) degrades dependences between neurons and across replicas. For large number of replicas, i.e., in the RMF limit, the neurons become asymptotically independent. Moreover, each neuron of type $i$ asymptotically receives inputs from neurons of type $j \neq i$ with Poissonian statistics across replicas. For often being only conjectured, the emergence of this simplified limit dynamics is referred to as the "Poisson Hypothesis" in network theory [48]. The Poisson Hypothesis was recently established rigorously for generic RMF limits, including LER models [51]. Under the Poisson Hypothesis, in RMF limits of LER networks, a neuron $i$ from a representative replica admits the effective dynamics given by

$$
x_i(t) = x_i(0) - \frac{1}{\tau_i} \int_0^t x_i(s)\, \mathrm{d}s + \sum_{j \neq i} \mu_{ij} P_j(t) - \int_0^t x_i(s^-)\, N_i(\mathrm{d}s)\,,
$$

where $\{P_j(t)\}_{j \neq i}$ denote independent Poisson processes with stationary firing rate $\beta_j$. Thus, at fixed network structure, the dynamics $x_i(t)$ only depends on the network background activity via the rates $\{\beta_j\}_{j \neq i}$. To be consistent under the assumption of stationarity, these rates must collectively satisfy the system of equations

$$
\beta_i = h_i \mathbb{E}[e^{a_i x_i}] = \mathcal{F}_i(\{\beta_j\}_{j \neq i})\,, \quad 1 \leq i \leq K\,,
\tag{10}
$$

where the notation $\mathcal{F}_i$ emphasizes that $\beta_i$ is evaluated as a function of the rates $\{\beta_j\}_{j \neq i}$. In the following, we will refer to $\mathcal{F}_i$ as a rate-transfer function. The fact that firing rates characterize alone the stationary distribution of the neural network is a feature of RMF limits. In fact, specifying RMF limits for LER networks consists in solving the self-consistent system Eq (10). This is one of the main goals of this work.

**Functional characterization via delay differential equations.** Heretofore, we have only considered RMF limits and their associated self-consistent system Eq (10) formally. To exploit RMF limits computationally, one must find explicit forms for the rate-transfer functions featured in Eq (10). Determining these explicit forms is the main technical challenge of the RMF approach and is the topic of the next section. Here, as a first step toward this goal, we establish functional relations between the stationary rates $\beta_i$ via probabilistic arguments. Specifically, we exploit the rate-conservation principle for point processes [66, 67] to exhibit a system of delay differential equations (DDEs) featuring the rates $\beta_i$.

In a nutshell, the rate-conservation principle states that in the stationary regime, any real-valued function of the network states reaches an equilibrium where there is a balance between its rate of increase and its rate of decrease. By virtue of the Poisson Hypothesis for RMF limits, we can apply the rate-conservation principle to any function of the internal states $x_i(t)$ independently. A natural choice is to consider exponential function $x_i(t) \mapsto e^{ux_i(t)}$, whose stationary expectation defines the moment-generating function (MGF) $L_i(u) = \mathbb{E}[e^{ux_i}]$. The MGF $L_i$ fully specifies the stationary distribution of $x_i(t)$ and is a well-behaved analytical function when it exists on an open interval containing zero. In the following, we assume that the MGF $L_i$ always exists in the RMF limit. By this, we mean that the MGF $L_i$ remains finite on an open interval containing zero, which implies analyticity in zero so that $x_i(t)$ admits moments of all order.

As a process, $t \mapsto e^{ux_i(t)}$ satisfies a stochastic equation which can be deduced from Eq (7) as

$$e^{ux_i(t)} - e^{ux_i(0)} = \quad -\frac{u}{\tau_i} \int_0^t x_i(s) e^{ux_i(s)} \mathrm{d}s$$

$$+ \sum_{j \neq i} (e^{u\mu_{ij}} - 1) \int_0^t e^{ux_i(s^-)} P_j(\mathrm{d}s) \qquad (11)$$

$$+ \int_0^t (1 - e^{ux_i(s^-)}) N_i(\mathrm{d}s) \,.$$

The three consecutive integral terms in the RHS correspond to continuous relaxation, independent Poisson bombardments, and post-spiking reset, respectively. At stationarity, we have $\mathbb{E}[e^{ux_i(t)}] = \mathbb{E}[e^{ux_i(0)}]$, so that the rate-conservation principle implies that the RHS has zero stationary expectation. In turn, we can interpret the stationary expectation of the three integrals in term of the MGF $L_i$ (see S1 Appendix for derivation). Ultimately, the rate-conservation principle takes the form of the following linear DDE

$$\frac{u}{\tau_i} L_i'(u) - V_i(u)L_i(u) - (\beta_i - h_i L_i(u + a_i)) = 0 \,, \qquad (12)$$

where we have introduced $V_i(u) = \sum_{j \neq i} \beta_j (e^{\mu_{ij}u} - 1)$ for brevity. In the above equation, the nonlocal term $L_i(u + a_i)$ is due to the exponential form of Eq (6) for the rate $\lambda_i(t)$ as interpreting the expected reset term in Eq (11) involves evaluating

$$\mathbb{E}[\lambda_i(s)e^{ux_i(s)}] = \mathbb{E}[e^{a_i x_i(s)} e^{ux_i(s)}] = L_i(u + a_i) \,. \qquad (13)$$

In the framework of perturbation theory, Eq (12) can be viewed as a singularly perturbed delay differential equation, whose perturbation parameter $h_i$, unlike many other more common cases, is on the delay term. To see this, note that $\beta_i = h_i \mathbb{E}[e^{a_i x_i}] = h_i L_i(a_i)$, so that Eq (12) reads

$$\frac{L_i'(u)}{\tau_i} - \frac{V_i(u)}{u}L_i(u) + h_i \left( \frac{L_i(u + a_i) - L_i(a_i)}{u} \right) = 0 \,, \qquad (14)$$

where $h_i$ appears as the coefficient of a forward discrete derivative. In the limit of $h_i \to 0$, Eq (14) becomes an analytically solvable, first-order homogeneous ODE. When $h > 0$, the presence of a delay term drastically changes the nature of the problem at stake, at least mathematically. Because of this drastic change, one needs to resort to techniques from singular perturbation theory to numerically solve Eq (14). In this regard, observe that Eq (14) also exhibits the relaxation rate $1/\tau_i$ as a singular perturbation parameter. Indeed, in the limit of $\tau_i \to \infty$, Eq (14) becomes a pure delay equation, with no relaxation component. However, by contrast with the ODE recovered when $h_i \to 0$, the delay equation obtained when $\tau_i \to \infty$ resists closed-form resolution. For this reason, $h_i$ will play the central part as a perturbation parameter.

The main caveat to solving Eq (14) for generic LER models ($1/\tau_i$, $h_i \neq 0$) is that the nonlocality of the DDEs Eq (12) precludes a direct analytical treatment, while posing problems with respect to solution uniqueness. To see this, observe that if one interprets the variable $u$ as a time parameter, the nonlocal term $L_i(u + a_i)$ formally introduces a negative delay $-a_i$. Due to the negativity of this delay, one can only solve the DDE Eq (14) of the form $x'(u) = f(x(u), x(u + a_i))$ backward, i.e., for decreasing value of $u$. Moreover, such solutions are only unique given the knowledge of some initial conditions on an interval of duration $a_i$. However, there is no natural notion of initial conditions at our disposal. Luckily, we will be able to address this caveat in a probabilistic setting. Specifically, we will see that there is a natural representation for a probabilistically interpretable solution to Eq (14), with no need to specify any initial conditions.

**Self-consistent equations via resolvent formalism.** Our goal is to characterize the unique MGF solution to the DDE Eq (14). Achieving this goal will allow us to give an explicit form to the rate-transfer functions formally defined by Eq (10). For simplicity, we omit all the neuronal indices whenever possible in the following. For instance, we will denote the output stationary firing rate of neuron $i$ by $\beta$ when unambiguous. Our strategy is to adapt the resolvent formalism to our delayed framework [68] in three steps.

First, we consider the forward discrete derivative term $(L(u + a) - L(a))/u$ as a known inhomogeneous term so that we can view Eq (14) as a linear ODE about $L$. The method of the variation of parameters yields integral forms for $L$ involving its shifted version $L(\cdot + a)$, but also some undetermined initial condition. We resolve the latter indeterminacy by selecting the only solution taking value $L(a) = \beta/h$, as required by the definition of the stationary rate: $\beta = \mathbb{E}[\lambda(t)] = h_i \mathbb{E}[e^{a_i x_i}] = hL(a)$. This yields the following integral equation

$$L(u) = \frac{\beta q(u)}{h} - h\tau \int_a^u \frac{q(u)}{q(v)} \frac{L(v + a) - L(a)}{v} \, dv \,, \tag{15}$$

where $q$ denotes the homogeneous solution to Eq (12) excluding the forward discrete derivative term: $q(u) = \exp\left(\tau \int_a^u V(v)/v \, dv\right)$.

Second, in view of Eq (15), we define the auxiliary function $H(u) = (L(u + a) - L(a))/u$. Substituting $H(u)$ into Eq (15), we obtain the following integral equation

$$H(u) = \frac{\beta}{h}\left(\frac{q(u + a) - 1}{u}\right) - q(u + a)\left[\frac{h\tau}{u}\int_a^{u+a} \frac{H(v)}{q(v)} \, dv\right], \tag{16}$$

where the delayed nature of the problem appears via the nonlocal integration upper bound. Observe that the inhomogeneous term in Eq (16) is analytic, whereas the integral term is analytic whenever $H$ is analytic. Eq (16) is the basis for adapting the resolvent formalism to our

delayed framework. To see this, let us define the sequence of iterated kernels

$$Q_m(u) = q(u + a)\left[\frac{1}{u}\int_a^{u+a}\frac{Q_{m-1}(v)}{q(v)}\,dv\right], \tag{17}$$

with $Q_0(u) = (q(u + a) - 1)/u$. Then, we can formally write a solution to Eq (16) as the series

$$H(u) = \frac{\beta}{h}\sum_{m=0}^{\infty}(-h\tau)^m Q_m(u)\,, \tag{18}$$

where $h$ appears as a perturbation parameter via the dimensionless quantity $h\tau$. Note that the kernels $Q_m$ are independent of $h$ but also of $\beta$, the yet-to-be-determined output firing rate. Note also that as the $Q_m$ are analytic in $u$, $H$ is an analytic function around zero as soon as the series converges uniformly on an open disk containing zero. We conjecture that this is always the case. This is supported by simulations which show that the probability density function of the internal variable $x$ is decaying superexponentially. Such superexponential decay implies that the moment generating function of $x$, as well as the function $H$, should be analytical (see S1 Fig).

Third, we specify $\beta$ as a function of the input parameters by exploiting the normalization constraint of the MGF: $L(0) = 1$. From the definition of $H$ above, we have that $-aH(-a) = L(0) - L(a) = 1 - \beta/h$. Thus, together with Eq (18), we must have

$$\frac{h}{\beta} = 1 - a\sum_{m=0}^{\infty}(-h\tau)^m Q_m(-a)\,, \tag{19}$$

where the dependencies on the input rates $\beta_j$, $j \neq i$, are mediated by the kernels $Q_m$. Observe that in the absence of inputs, i.e., for $V(u) = 0$, we have $q(u) = 1$ so that all the terms $Q_m(-a) = 0$. Thus, in the absence of inputs, we recover consistently that $\beta = h$, the spontaneous spiking rate. In the presence of inputs, Eq (19) determines the rate-transfer function $\mathcal{F}$ of a neuron in a feedforward network. When considering a recurrent network, these rate-transfer functions define the sought-after self-consistency equations Eq (10), which must be jointly satisfied. These equations take the explicit forms:

$$\beta_i = h_i\left[1 - a_i\sum_{m=0}^{\infty}(-h_i\tau_i)^m Q_{i,m}(-a_i; \{\beta_j\}_{j\neq i})\right]^{-1}\,, \tag{20}$$

where we have indexed all neuron-specific quantities and highlighted the dependencies on the input rates $\beta_j$, $j \neq i$. In the following, we will refer to the quantities computed by Eq (20) as "RMF calculation". We discuss the numerical methods used to perform the RMF calculation in the next section.

## Numerical methods

This section accounts for the numerical methods used to estimate the rate-transfer functions formally expressed by Eq (19). This section is not required to read the ensuing results and discussion sections.

### Divergent perturbative series

The formal series expansion Eq (19) suggests a natural numerical scheme to compute $\beta$. At stake is to compute the values $Q_m(-a)$, $m \geq 0$, in order to approximate the output rate $\beta$ by

truncating the series $\sum_{m=0}^{\infty} Q_m(-a)y^m$ at $y = -h\tau$. As the functions $Q_m$ are defined iteratively via the nonlocal integral relation given in Eq (17), the numerical computation of $Q_m(-a)$ requires mesh grids in different ranges for different $m$. Specifically, calculating $Q_m(-a)$ requires the knowledge of $Q_{m-1}$ on the domain $[0, a]$. Knowing $Q_{m-1}$ in the domain $[0, a]$ requires the knowledge of $Q_{m-2}$ on the domain $[0, 2a]$. By straightforward iteration, calculating $Q_m(-a)$ ultimately requires all the $Q_n$, $0 \leq n \leq m - 2$, to be known in the range $[0, (m - n)a]$, respectively. Therefore, denoting the ultimate order of the approximation by $M$, we first evaluate $Q_0$, which we know in closed form, on some mesh grid on the domain $[0, Ma]$. Then, we proceed to sequentially evaluate $Q_{n+1}$ for increasing order by numerical integration of $Q_n$ via Eq (19) over the domain $[0, (M - n)a]$. After repeating this iteration $M$ times, we collect $Q_0(-a)$, $Q_1(-a)$, $\cdots$, $Q_M(-a)$. We will discuss how to choose the approximation order $M$ in the next section.

In principle, one can hope to compute $\beta$ by direct summation of the Taylor series involving the collected coefficients $\{Q_m(-a)\}_{m=0}^{M}$. The convergence of such a series would be justified within the resolvent formalism if the functional map $Q_m \mapsto Q_{m+1}$ given by Eq (17) is a contraction [69]. Unfortunately, this condition does not hold for moderately large excitation as shown in Fig 2, which estimates the radius of convergence of the series as $r = \lim_{m \to \infty} r_m$ with $1/r_m = |Q_m(-a)|^{1/m}$. Fig 2a shows that $1/r_m$ grows superexponentially when the neuron receives excitatory inputs. This indicates a zero radius of convergence so that direct summation using Eq (19) will fail whenever $h \neq 0$. By contrast, Fig 2b shows that $1/r_m$ admits a finite limit when the neuron is subjected to inhibitory inputs alone. This is also true for weak inputs, as shown in Fig 2c and 2d.

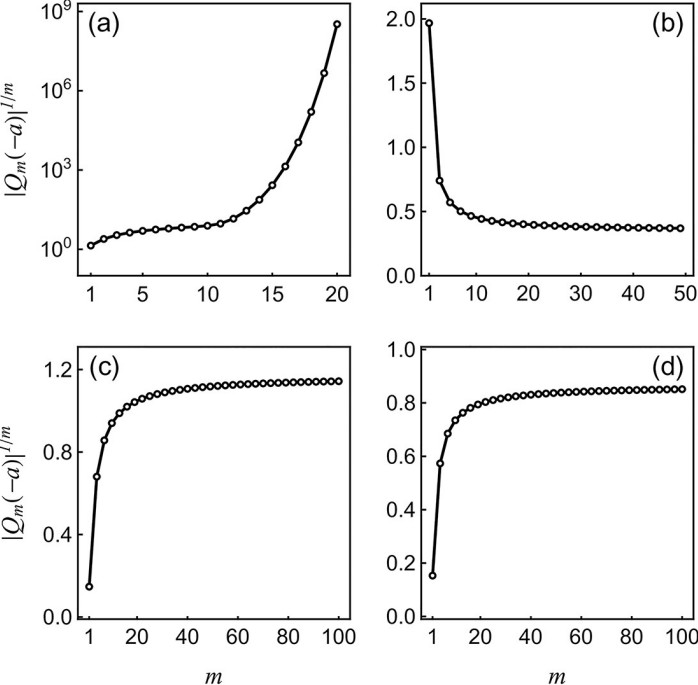

**Fig 2. Plots of $1/r_m = |Q_m(-a)|^{1/m}$ v.s. $m$.** This figure shows the numerical values of the series coefficients subjected to different types of inputs. The input rates $\beta_{e,i} = 0.5$ kHz are the same across all four panels. In panel (a), the neuron is subjected to an excitatory input with $\mu_e = 3$. The superexponential growth of $1/r_m$ is clearly manifested in this log-scale plot. In panel (b), the neuron is subjected to an inhibitory input with $\mu_i = -3$. In panel (c) and (d), the neuron is subjected to a weak excitatory and inhibitory input with $\mu_e = 0.3$ and $\mu_i = -0.3$ respectively. In these three cases, $1/r_m$ converges to finite values. Parameters: $h = 1$ Hz, $a = 0.1$, $\tau = 10$ ms.

This numerical evidence suggests that $\beta$, as a function of the perturbation parameter $h$, is nonanalytic in zero whenever the neuron is driven by strong excitatory inputs. Moreover, even when the series has finite radius, we will see that the convergence is only possible for a very restricted range of inputs (see the next section). This is due to $h$ being a singular perturbation parameter when $h \rightarrow 0$. There exist different techniques to approximate the general solution of singular perturbation problems such as, e.g., multi-scale analysis [70] or matched asymptotics of distinguished solutions [71]. Here, we leverage the knowledge of the divergent series to form convergent approximations via Padé theory [46, 47].

## Padé approximants

Direct Taylor series summation fails for neurons whose drive is dominated by excitation. This is a major drawback as nonlinear network dynamics typically involve regimes in which subsets of neurons are strongly excited. Such regimes will typically arise in the RMF limits of metastable systems, for which active groups of neurons are transiently stabilized by strong recurrent excitation. To account for strong excitation drives, we need to improve the domain of convergence of our numerical methods. This can be done by evaluating rate-transfer functions via Padé approximants summation [46, 47].

Formally correct but numerically divergent series stem from "invalid" Taylor expansions because the point of evaluation lies outside the radius of convergence or because the function is singular at that point in the first place, with a zero radius of convergence. Intuitively, divergent series result from trying to approximate a function with nonanalytic singularities by polynomial approximations of increasing degree, whose limit behavior is bound to be analytic. To address this point, Padé approximants substitute polynomials for rational functions. Informally, the use of rational functions allows for a better approximation of nonanalytic functions by mimicking their singularities via the poles of the approximants. Given a function $f(x)$, its $[m, n]$ Padé approximant at $x = 0$ is the ratio of a degree-$m$ polynomial and a degree-$n$ polynomial:

$$[m, n](x) = \frac{p_0 + p_1 x + p_2 x^2 + \cdots + p_m x^m}{1 + q_1 x + q_2 x^2 + \cdots + q_n x^n}, \tag{21}$$

where the $m + n + 1$ coefficients are determined so that $f(x) - [m, n](x) = O(x^{m+n+1})$, i.e., the Taylor expansion of $f(x)$ agrees with that of $[m, n](x)$ up to the first $m + n + 1$ terms. A common approximating scheme is given by the zigzag diagonal chain:

$$\cdots \rightarrow [m, m] \rightarrow [m, m+1] \rightarrow [m+1, m+1] \rightarrow \cdots .$$

For fixed $x$, if the Padé approximants along this chain converge, we assign the limiting value to be the functional value of the original divergent series. This leads us to the numerical criteria for determining the cut-off order $M$: we choose $M$ to be the smallest integer such that $|[M/2, M/2](x) - [M/2 - 1, M/2](x)| < \delta$, given an error level $\delta > 0$.

The parametric range for which the Padé approximants summation converges to the correct result generally exceeds that of the direct Taylor series summation. To check this, we consider a single neuron subjected to an excitatory input of varying rate $\beta_e$ and strength $\mu_e$. For each parametric point $(\beta_e, \mu_e)$, we test if the computed output rate $\beta$ lies within one standard deviation of the simulated $\beta$. The results are shown in Fig 3, where each panel corresponds to a different set of $h$ and $a$. From Fig 3a, we observe that the direct method is only accurate for weak excitatory inputs. The contour curve of the convergent parametric regime is approximately linear in this log-scale plot, which means the maximal computable excitatory input rate $\beta_e$ is exponentially decreasing when we increase the excitatory synaptic strength $\mu_e$. By

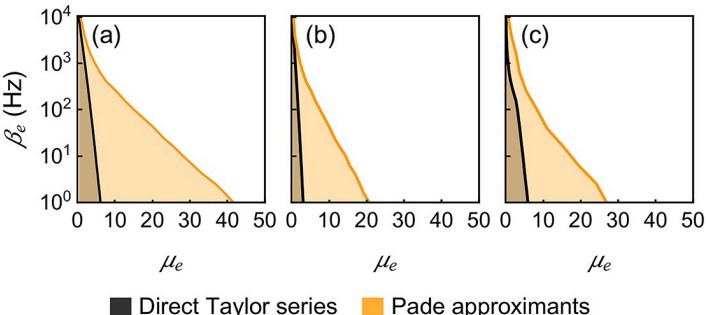

**Fig 3. Convergent parametric regimes.** These figures show the comparisons of the convergent parametric regimes for the direct Taylor series summation and Padé approximants summation. The convergence regimes are shaded in different colors and their envelope curves are shown. The vertical axis is in log scale. The convergence of each parametric point $(\beta_e, \mu_e)$ is determined by whether or not the computed $\beta$ value is within one standard deviation of the simulated $\beta$ value. In panel (a), $h = 1$ Hz, $a = 0.1$; in panel (b) $h = 1$ Hz, $a = 0.2$; in panel (c) $h = 50$ Hz, $a = 0.1$. Parameters: $\tau = 10$ ms.

contrast, the Padé approximant summation yields a significantly better coverage on the $(\beta_e, \mu_e)$ parametric space. Although the maximal $\beta_e$ still decreases exponentially when $\mu_e$ is large, the decay is much slower, especially in the low and moderately high rate regime. Fig 3b shows the convergent parametric regimes with $a = 0.2$ doubled from $a = 0.1$. Both of the regimes shrink into half, which is expected considering that in Eq (6), only the product of $a$ and $\mu$ matters. The nonlinear dependence of the convergent regime is more apparent when $h$ becomes large. As shown in Fig 3c, when $h = 50$ Hz, the convergent regime of the direct Taylor summation in the high $\beta_e$ regime is highly suppressed compared with that of Padé approximants.

## Computational efficiency

The RMF approach consists in approximating the exact dynamics of finite-size networks by the computationally tractable dynamics of the corresponding RMF limits. As infinite networks, these RMF limits are not amenable to exact simulations. This is by contrast with the finite network of interest, whose dynamics can be simulated exactly via Monte-Carlo methods [32]. Fortunately, RMF dynamics are entirely characterized by the stationary rates solving the self-consistent equations (20). The RMF computational efficiency is measured by the numerical cost of solving these equations.

To assess the RMF computational efficiency, we compare the numerical cost of solving the self-consistent equations (20) with that of estimating exact stationary rates via Monte-Carlo methods. For convenience, we perform such a comparison in the single neuron setting, where the RMF approximation is exact. Specifically, we consider a single neuron subjected to excitatory inputs of size $\mu_e = 1$ delivered at $\beta_e = 1$kHz. For such parameters, our RMF calculations converge to finite values. We compare these rate values with Monte-Carlo estimates obtained via an exact event-driven simulation scheme detailed in S1 Appendix. In all generality, the precision of these latter estimates depends on the number of simulated spiking events. For this reason, we perform Monte-Carlo simulations for a fixed number of spiking events ranging from 10 to $10^4$. Then, we compute the mean and standard deviation of the rate estimates by averaging over 32 independent repeats.

In Fig 4a, we compare Monte-Carlo estimates (circles) obtained for a varying number of spiking events with the corresponding RMF estimates (dashed line). As expected, the simulation estimates and RMF calculations agree but the standard deviation of simulation estimates gets larger when computed for fewer spiking events. In Fig 4b, we plot the relative standard

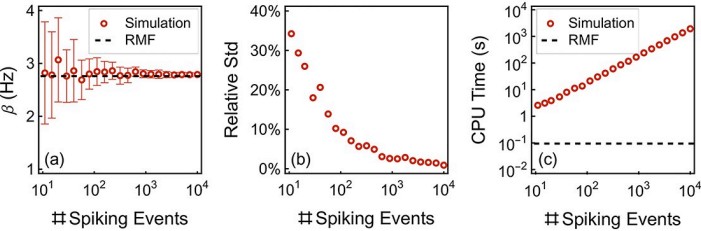

**Fig 4. Computational efficiency of the RMF method.** Panel (a) shows the spiking rates of a single neuron subjected to a fixed excitatory input with
$\beta_e = 1$ kHz and $\mu_e = 1$. The black dashed line indicates the RMF calculations of the rate. The red circles and error bars indicate the simulated mean and standard deviation of the spiking rate computed from different numbers of spiking events. Specifically, we run the event-driven simulation until $N$ spiking events for the single neuron are accumulated. This process is repeated for 32 times. During each repetition, we record the total time $T_k$ ($k = 1, \cdots, 32$). The mean firing rates of the neuron are then computed by $\bar{\beta} = (\sum_{k=1}^{32} N/T_k)/32$. The standard deviations of the rates (indicated by error bars) over these 32 repetitions are computed by taking the square root of $(\sum_{k=1}^{32} (S = N/T_k)^2 - \bar{\beta}^2)/(32 - 1)$. Panel (b) shows the relative standard deviations of the simulated rates. The relative standard deviations are computed by $\mathrm{Std}(\beta)/\bar{\beta}$. Panel (c) shows the CPU time consumed for simulating different numbers of spiking events with 32 repetitions (red circles), together with the CPU time of the RMF calculation (black dashed line). The simulation platform is a 2021 MacBook Pro with Apple M1 Pro chip. Parameters: $\tau = 10$ ms, $h = 1$ Hz, $a = 0.1$.

deviation of the simulated estimates. We find that to achieve moderately accurate simulation estimates, e.g., within 5% relative standard deviation, we need about 400 spiking events per repeat, i.e., about $1.3 \times 10^5$ inputs. In Fig 4c, we plot the corresponding processing times for both methods. As expected, the time used by Monte-Carlo simulation scales linearly with the desired number of spiking events. Achieving moderately accurate simulation estimates takes several minutes of processing time, whereas our RMF calculation takes a comparably negligible amount of processing time (0.09s in this example).

## Results

### Rate-transfer functions

**Nonlinear corrections and reset mechanism.**   The derivation of Eq (14) shows that its delayed nature is due to the presence of a post-spiking reset mechanism. This post-spiking reset mechanism becomes irrelevant when $h = 0$ as Eq (14) simplifies to the solvable homogeneous ODE: $uL'(u) = V(u)L(u)$. Then, the normalization condition $L(0) = 1$ imposes that $L(u) = q(u)/q(0)$, leading to $\beta = hL(a) = h/q(0)$. The abrupt loss of the delayed terms in Eq (12) for $h = 0$ reveals the latter solution as a distinguished limit for $L$ when $h \to 0$. In that respect, one can check that $h/q(0)$ is precisely the first-order term in Eq (20) and reads explicitly

$$\beta = he^{\sum_j \beta_j w_j} + O(h^2) \quad \text{with} \quad w_j = \tau \int_0^a (e^{\mu_j v} - 1)/v \, dv, \qquad (22)$$

where $w_j$ is the effective synaptic weight of upstream neuron $j$. The nontrivial dependence of the effective weights $w_j$ on the original synaptic strengths $\mu_j$ follows from including finite-size effects in the RMF framework [28, 52]. For small synaptic weights $\mu_j \ll 1/a$, we recover the classical mean-field regime for which the linear scaling $w_j \simeq \tau\mu_j a$ holds. We will later see that even for moderate values of $\mu_j$, finite-size effects can have a substantial impact on evaluating the stationary moments of the dynamics.

Eq (22) reveals that any nonlinear correction to the first-order estimate is due to including the reset mechanism. For Eq (20) being a singular expansion, these higher-order corrections do not always yield a convergent series. However, we expect the first-order term to be a valid

approximation as long as the reset timescale, i.e., the mean interspike interval $1/\beta$, remains large compared with the relaxation time $\tau$: $\tau \ll 1/\beta$. This is because in this regime, fast relaxation erases the slower influence of the reset mechanism in shaping the distribution of the internal variable $x$. By contrast, in the regime of moderately large excitation $1/\beta \simeq \tau$, the reset mechanism starts to significantly impact the dynamics of $x$ and to dampen the first-order exponential dependence of $\beta$ on the input rates $\beta_j$. In principle, neurons can fire at a rate up to 200Hz for a leak time constant $\tau$ larger than 10ms. Thus, if one interprets the internal variable $x$ as a proxy for the membrane voltage, the biophysically relevant range of neural activity includes values in the intermediary regime $\beta\tau \simeq 1$, for which nonlinear corrections are necessary. Exploring this regime requires using the Padé approximants associated to the divergent series.

We confirm numerically the above discussion by considering the simplest excitatory neuronal model, whereby a neuron is subjected to an input of rate $\beta_e$ via the positive synaptic weight $\mu_e$. Fig 5a and 5b quantify nonlinear corrections to the first-order prediction by plotting $\beta/h$ as a function of $h$ at fixed input conditions. As expected, the no-reset limit $\beta = h/q(0)$ holds for $h \to 0$. By contrast, in the opposite limit $h \to \infty$, the frequent resets due to spontaneous spiking erase the impacts of the relaxation as well as the inputs in between spikes, so that the spontaneous spiking emission dominates the dynamics: $\beta \to h$. By comparison with Fig 5a and 5b shows that the stronger excitatory drive, the larger the correction to the first-order terms. We mark the transition between the two asymptotic regimes by the mid-point $h_{1/2}$ such that $\beta(h_{1/2}) = h(1 + 1/q(0))/2$. To explore the domain of validity of the first-order expansion, we then check that at the mid-point value $h_{1/2}$, we have $\beta\tau \simeq 1$, even for eventually large nonlinear correction. To this end, we perform a systematic RMF calculation to represent $\beta\tau$ for $h_{1/2}$ as a function of the input rate $\beta_e$ and the synaptic weight $\mu_e$. Fig 5c shows that the stronger the input, the smaller value of $\beta\tau$. However, this dependence is weak and $\beta\tau \simeq 1$ holds throughout. This confirms our analysis about the domain of validity of the first-order approximation.

**Moment analysis of the neuronal response.** The crux of the RMF approach is to capture the stationary dynamics of a neuron via a parametric probabilistic model. Moreover, this model admits the output rate $\beta$ as a sufficient statistics, assuming the biophysical parameters

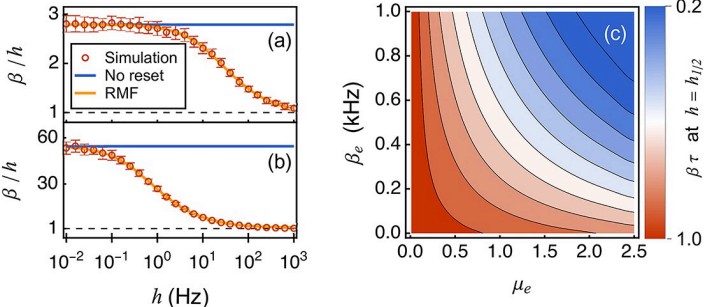

**Fig 5. Nonlinearity of firing rates with post-spiking reset mechanism.** In panel (a) and (b), the log-linear plots of $\beta/h$ v.s. $h$ show the nonlinearity resulted from the post-spiking reset mechanism. In panel (a), the neuron is subjected to a single excitatory input with $\beta_e = 1$ kHz and $\mu_e = 1$, while in panel (b), the input is of $\beta_e = 1.5$ kHz and $\mu_e = 2.5$. Panel (c) is the contour plot of the value of $\beta\tau$ at mid-point $h_{1/2}$, where excitatory input rate $\beta_e$ and strength $\mu_e$ are varied. The contours are equidistant with respect to the value of $\beta\tau$. For the simulated data, we run the event-driven simulation until 400 spiking events for the single neuron are accumulated. This process is repeated 32 times. During each repetition, we record the total time $T_k (k = 1, \cdots, 32)$. The mean firing rates of the neuron are then computed by $\bar{\beta} = (\sum_{k=1}^{32} 400/T_k)/32$. The standard deviations of the rates (indicated by error bars) over these 32 repetitions are computed by taking the square root of $(\sum_{k=1}^{32} (S = 400/T_k)^2 - \bar{\beta}^2)/(32 - 1)$. Parameters: $a = 0.1$, $\tau = 10$ ms.

and the input rates known. This means that given these assumptions, all the moments of the stationary neuronal dynamics can be deduced from knowing $\beta$. For LER neurons, the moments of interest are those of the stochastic intensity, $M_n(\lambda) = \mathbb{E}[\lambda^n]$, $n > 1$, and those of the internal variable $x$, $M_n(x) = \mathbb{E}[x^n]$, $n > 0$. With knowledge of $\beta$, $M_n(x)$ can also be computed efficiently for all $n > 0$ via expansions akin to Eq (20) (see S1 Appendix). In turn, this allows one to estimate $M_n(\lambda)$, $n > 1$, by truncation of the series

$$M_n(\lambda) = h^n \sum_{k=0}^{\infty} \frac{(an)^k}{k!} M_k(x) \,. \tag{23}$$

Here, we demonstrate that the above RMF computational framework accurately quantifies the response of LER neurons subjected to Poissonian bombardments. We proceed by comparison with exact, event-driven, Monte-Carlo simulations under various driving conditions and for biophysically-relevant parameter values [33, 34]. Bear in mind that in all cases, our computational results are obtained incomparably faster than those estimated via Monte-Carlo simulations. We will see that such a computational advantage leverages to neural networks in the next section. In demonstrating our point, we will discuss the general features of the LER neuronal response in light of the competition existing between the two timescales at play: the relaxation time $\tau$, and the mean interspike interval $1/\beta$. We will also consider closed-form approximations for various regimes of activity.

For a feedforward neuron, the input-rate-dependence of $\beta$ is encoded via the rate-transfer function Eq (20). In Fig 6, we explore this rate-transfer function numerically to reveal that—perhaps surprisingly—LER neurons essentially behave as stochastic rectifier linear units (ReLUs). In addition to our RMF calculations, we consider three types of approximations for comparison. At a low spiking rate, the relaxation timescale dominates, e.g., $\tau \ll 1/\beta < 1/h$, and we utilize the first-order, no-reset approximation. At a high spiking rate, e.g., $\tau \simeq 1/\beta$, we consider two heuristically-derived approximations in the TMF limit and with or without relaxation. Both approximations are obtained via a simple probabilistic argument (see S1 Appendix).

In Fig 6a, when excitation dominates, the zero-order exponential approximation breaks down when $\beta$ exceeds 10Hz, which happens at about 2kHz for the considered synaptic strength. Note that the input rates quantify the frequencies of synaptic activations so that 2kHz corresponds to, e.g., 20 upstream neurons firing at 100Hz or 200 upstream neurons firing at 10Hz. By contrast, the RMF calculations accurately predict the quasilinear input-rate dependence of $\beta$ (up to convergence failure). At high drive $\beta_e$, the output rate adjusts so that on average, the $\beta_e/\beta$ received inputs are canceled by a single reset over the timescale $1/\beta$. This balance generically leads to a weakly sublinear rate dependence in the RMF limit as well as in the TMF limit. In the TMF limit, we heuristically establish that at large $\beta_e$, $\beta_{\mathrm{TMF}} \simeq a\beta_e \, \mu_e/\ln(1 + a\beta_e \, \mu_e/h)$. This TMF approximation produces the right scaling but only becomes accurate for exceedingly high rates, when finite-size effects become negligible (see inset for the ratio $\beta/\beta_{\mathrm{TMF}}$). In Fig 6b, we show that the mean internal variable $\bar{x} = M_1(x)$ mirrors the behavior of $\beta$ after logarithmic compression. A low-rate linear growth $\bar{x} \simeq \tau\beta_e\mu_e$ is followed by a logarithmic behavior $\bar{x} \simeq \ln(1 + a\beta_e\mu_e/h)/(2a)$ at high drive. In Fig 6c and 6d, when inhibition dominates, the neuron seldom spikes so that $\tau \ll 1/\beta$, and the reset becomes irrelevant. Then, long interspike intervals allow for the integration of many inputs so that finite-size effect can be neglected as in the heuristic TMF limit. As a result, all approximations perform accurately.

Fig 7 compares the second-moment predictions deduced from our various approximations. Comparing Fig 7a and 7b with Fig 7c and 7d consistently shows that predicting the variability in both $\lambda$ and $x$ requires RMF calculation when excitation dominates whereas the zero-order, no-reset approximation suffices when inhibition dominates (see S1 Appendix). Aside from

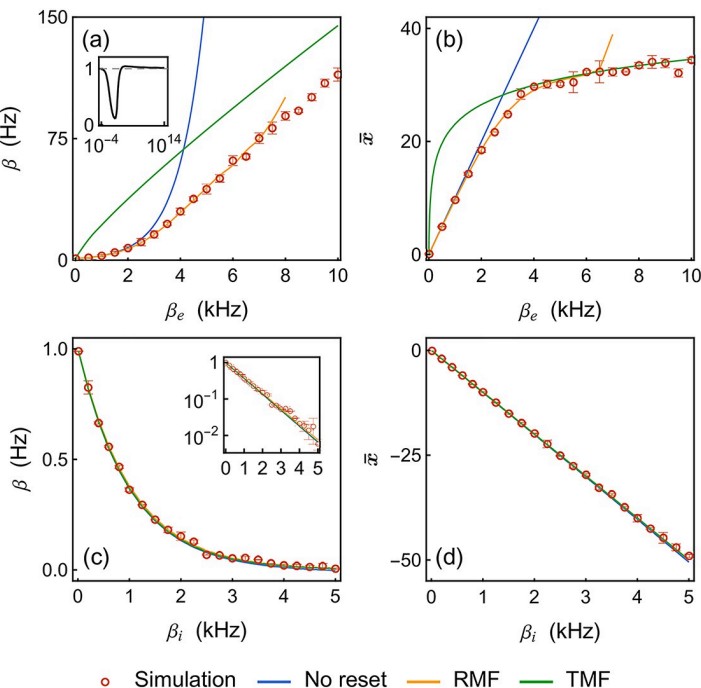

**Fig 6. Rate-transfer functions: Mean firing rates $\beta$ and mean internal variables $\bar{x}$.** For panel (a) and (b), the neuron is subjected to an excitatory input with varying rates $\beta_e$ from 0 to 10 kHz and fixed strength $\mu_e = 1.0$. In panel (a), the top-left inset shows the ratio $\beta/\beta_{\mathrm{TMF}}$ in a wider range of input rates. For panel (c) and (d), the neuron is subjected to an inhibitory input with varying rates $\beta_i$ from 0 to 5 kHz and fixed strength $\mu_i = -1.0$. The inset in panel (c) is in logarithmic scale to show its exponential dependence. For the simulated data, we run the event-driven simulation until 400 spiking events for the single neuron are accumulated. This process is repeated 32 times. During each repetition, we record the total time $T_k(k = 1, \cdots, 32)$. The mean firing rates of the neuron are then computed by $\bar{\beta} = \left(\sum_{k=1}^{32} 400/T_k\right)/32$. The standard deviations of the rates (indicated by error bars) over these 32 repetitions are computed by taking the square root of $\left(\sum_{k=1}^{32} \left(S = 400/T_k\right)^2 - \bar{\beta}^2\right)/(32 - 1)$. Meanwhile, we record the entire time series of the internal variable $x$, with which we compute the mean of $x$. The standard deviations of the internal variable $x$ (indicated by error bars) are computed over these 32 repetitions. Parameters: $h = 1$ Hz, $a = 0.1$, $\tau = 10$ ms.

this core observation, two remarks are worth making: First, we remark in Fig 7a and 7b that the RMF calculations interpolate well between the low-rate and high-rate variability regimes. When the input rate is low, the variability mainly comes from the relaxation as in the no-reset limit. When the neuron spikes more frequently in response to strong drive, the reset mechanism becomes the dominant source of variability. The latter can be captured asymptotically by the TMF limit. Second, we remark in Fig 7c and 7d that the TMF approximations fail to capture neural variability for both quantities, even though the mean response is accurately predicted in the inhibitory case. This is because TMF limits inherently neglect finite-size effects by assuming that the stochastic intensity varies deterministically in between spikes. Such an assumption leads to drastic underestimation of the variability in the inhibition-dominated regime, at least before the neuron becomes virtually silent.

**Finite-size effects and balanced regime.** As mentioned above, a benefit of the RMF framework is that it enables the study of finite-size effects. In TMF approximations, finite-size effects are erased in the process of scaling interactions in the limit of infinite-size networks. As a result of this process, synaptic weights only appear as multipliers of the input rates in the rate-transfer functions [23]. This is not the case in RMF limits as shown by the nontrivial dependence of the effective weights $w_j$, which act as rate multipliers, on the original weight $\mu_j$ in Eq (22).

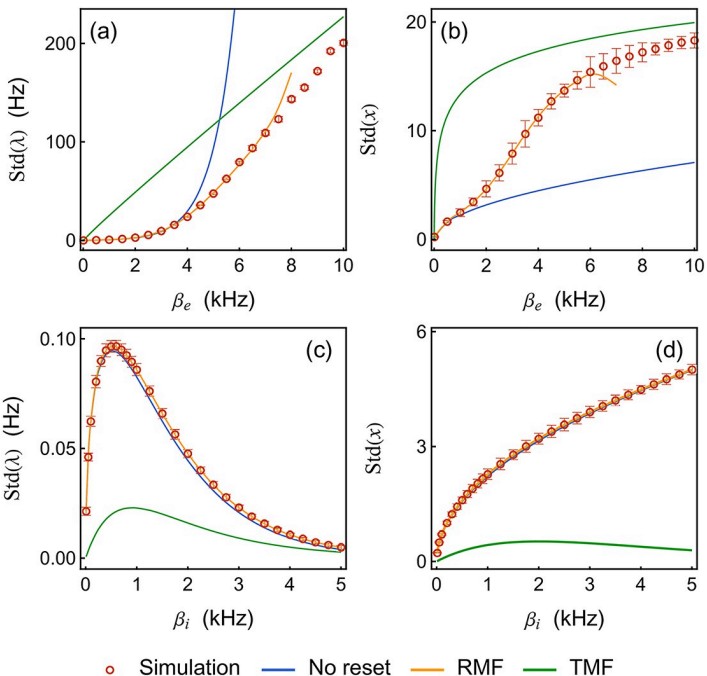

● Simulation — No reset — RMF — TMF

**Fig 7. Rate-transfer functions: Standard deviation of the neuron firing rates Std($\lambda$) and of the internal variables Std($x$).** For panel (a) and (b), the neuron is subjected to an excitatory input with varying rates $\beta_e$ from 0 to 10 kHz and fixed strength $\mu_e$ = 1.0. For panel (c) and (d), the neuron is subjected to an inhibitory input with varying rates $\beta_i$ from 0 to 5 kHz and fixed strength $\mu_i$ = −1.0. For the simulated data, we run the event-driven simulation until 400 spiking events for the single neuron are accumulated. This process is repeated 32 times. During each repetition, we record the entire time series of the internal variable $x$, with which we can compute the moment of $x$ for any given order. Std($\lambda$) is computed from Eq (23) with cut-off order 15. Std($x$) is computed by taking the square root of $M_2(x) - M_1^2(x)$. The standard deviations of the quantities (indicated by error bars) are computed over these 32 repetitions. Parameters: $h$ = 1 Hz, $a$ = 0.1, $\tau$ = 10 ms.

We quantify these finite-size effects by elucidating the dependence of the mean and variance of the LER neuronal response on the synaptic weights at fixed overall mean drive. By this, we mean that we jointly vary the numbers $K_e$, $K_i$ and the weights $\mu_e$, $\mu_i$ of the synapses so as to maintain the overall levels of excitation and inhibition, denoted by $E = K_e \mu_e$ and $I = K_i \mu_i$. Specifically, we compare the three following RMF approximations with their TMF counterparts: (1) $K_e$ excitatory inputs alone; (2) $K_i$ inhibitory inputs alone; (3) $K_e$ excitatory inputs balanced by $K_i$ inhibitory inputs with $K_e = K_i$ and $E = I$. The latter balance condition is a core assumption to a broad class of models accounting for the maintenance of neural variability in the limit of infinite-size networks, the so-called balanced network models [72].

Fig 8 shows the RMF calculations and the simulated values for the mean and standard deviation of the internal variable with biologically relevant parameters. In all cases, our RMF predictions coincide with simulated results to numerical error. As expected for excitatory inputs (Fig 8a and 8a'), the input variability tends to average out for large number of inputs $K_e$. Accordingly, we observe that the TMF limit is accurate in this regime. By contrast, when $K_e$ is small and finite-size effects are no longer negligible, the TMF limit marginally overestimates the mean, while underestimating the standard deviation by about twofold when $K_e$ = 7. This is consistent with the TMF limit erasing variability, even in the presence of a stochastic reset. When driven by purely inhibitory inputs (Fig 8b and 8b'), neurons remain silent for long period over which the variability in the inputs averages out. As a result, the TMF limit

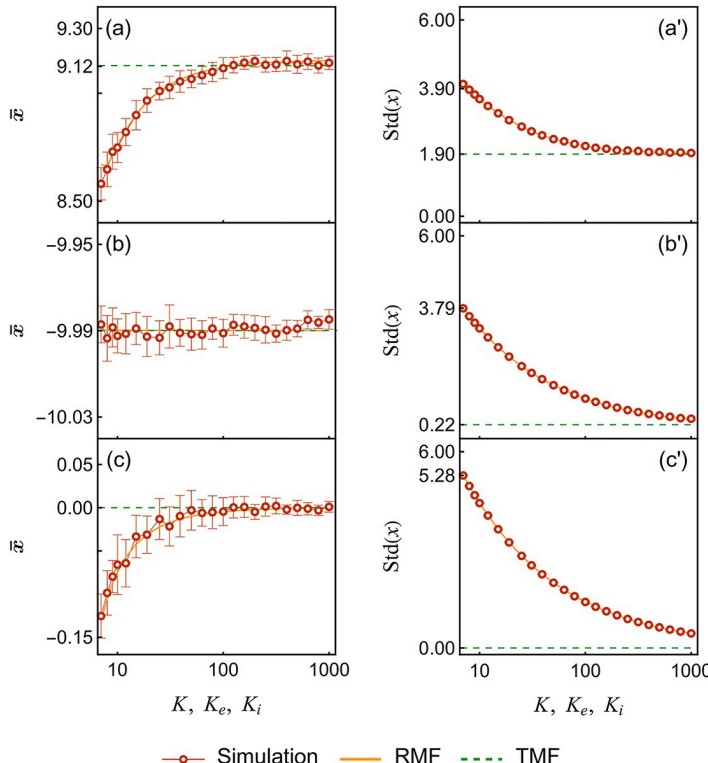

**Fig 8. Finite-size effect.** This figure shows the means and standard deviations of the internal variable of a neuron subjected to different configurations of input, varying the number of input channels. For all panels, $E = 20$ and $I = -20$, $\mu_e = E/K_e$, $\mu_i = I/K_i$. Panel (a) and (a'): excitation only, $\beta_e = 50$ Hz. Panel (b) and (b'): inhibition only, $\beta_i = 50$ Hz. Panel (c) and (c'): $K_e = K_i = K$, $\beta_e = \beta_i = 50$ Hz. For the simulated data, we use the same methods described in the captions of Figs 6 and 7 to compute the values (points) and standard deviations (error bars). Parameters: $a = \ln(100)/20 \approx 0.23$, $h = 1$ Hz, $\tau = 10$ ms.

performs well for all input number $K_i$. Nevertheless, the neural variability is still largely underestimated. This is again because TMF generally erases input variability, with a more drastic effect with inhibitory inputs. For instance, the TMF limit yields a seventeenfold reduction of the variability for $K_i = 7$. The overperformance of RMF over TMF is magnified in the balanced case (Fig 8c and 8c'). Under this condition, the TMF limit gives zero mean and variance for the internal variable as there is no net drive: $\sum_j \beta_j \mu_j = 0$. However, the corresponding finite-size system is dominated by the next order fluctuation, yielding nonzero mean and variability. Actually, one can check that for balanced conditions, the mean firing rate increases in keeping with the standard deviation of $x$, and against the mean of $x$, as $K_e = K_i$ decreases. This confirms that the neural response is dominated by input variability in this regime. The slight negative trend of the mean of $x$ is due to the nonlinearity of the exponential function, which biases against upward excursions from zero.

## Metastability in the RMF limit

**Network activity via fixed-point resolution scheme.** We are now in a position to characterize the dynamics of a recurrent LER network in the RMF limits. Recall that such limit dynamics are approximate versions of the original $K$-neuron dynamics, which are obtained via randomization of interactions across an infinite number of replicas. As a result of this

randomization, within each replica, each neuron $i$ experiences inputs as if delivered according to independent Poisson processes with rate $\beta_j$, $j \neq i$. The parametrization of the inputs via rates alone is the root cause for the RMF frameworks being computationally tractable. Within a recurrent network, these rates need to be determined by solving the system Eq (20) for self-consistent stationary rates $\boldsymbol{\beta} = \{\beta_1, \ldots, \beta_K\}$. By virtue of its interpretation in terms of a physical limit, the RMF system Eq (20) must admit some solutions $\beta^\star$. These solutions can be computed efficiently for LER networks.

Interpreting Eq (20) as a fixed-point problem naturally leads to estimate possible solutions $\beta^\star$ via the naive iterative scheme: $\boldsymbol{\beta}_{n+1} = F(\boldsymbol{\beta}_n)$. Aside from issues of Padé convergence, we expect this naive scheme to be locally convergent because the rate-transfer functions are weakly sublinear for large rates, which precludes runaway iterations. At the same time, these rate-transfer functions are also strongly supralinear (exponential) at low rate, so that solution uniqueness is not necessarily guaranteed. In practice, we find that whenever the Padé approximants summation converges, the naive scheme always locally converges toward a solution $\boldsymbol{\beta}_n \to \boldsymbol{\beta}^\star$, which may depend on the initial condition $\boldsymbol{\beta}_0$. A good choice for the initial condition is of the form $\boldsymbol{\beta}_0 = \boldsymbol{h} + \epsilon$ where $\epsilon$ represents a possibly random perturbation. Such initial conditions can be made to fall into the region of Padé convergence, while it is possible to achieve distinct solutions by tuning $\epsilon$ (if several solutions exist). We compare the RMF and simulated rates for two network structures with strong connections: In Fig 9a and 9b, the considered network has a dominant sparse feedforward structure, which promotes input independence as in the RMF limit. Accordingly, RMF rate approximations yield accurate predictions. In Fig 9c and 9d, the considered network has a dense recurrent structure, which promotes input correlations in excitatory networks. However, recurrent inhibition appears to maintain input independence and RMF calculation is accurate as well.

The two networks considered above have a single fixed-point RMF solution $\boldsymbol{\beta}^\star$. Intuitively, this is because the original system has a single "equilibrium" state, so that the corresponding ergodic dynamics is dominated by a single relaxation time. By contrast, metastable systems admit several local "pseudo-equilibria", leading to multi-timescale dynamics: At small

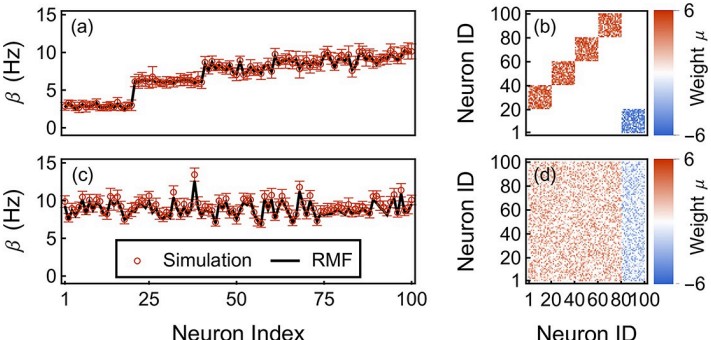

**Fig 9. Comparison of RMF and simulated rates in inhomogeneous networks.** Two networks of 80 excitatory and 20 inhibitory neurons are considered. In panel (a), the network is feedforward with 4 excitatory layers and one last inhibitory layer inhibiting the first layer. The neurons between layers are connected with probability 75% and the synaptic strengths are uniformly random between 3 and 6, as shown in (b). Panel (c) and (d) are for a network with no particular structure. Each pair of neurons is connected with probability 50% and the synaptic strengths are uniformly random between 0 and 4. For the simulated data, we run the event-driven simulation until 40000 spiking events for the entire network are accumulated. This process is repeated 32 times. During each repetition, we record the total time $T_k (k = 1, \cdots, 32)$ and spiking event count $S_{i,k}$ for each neuron $i (i = 1, \cdots, 100)$. The mean firing rates of neurons are computed by $\bar{\beta}_i = \left(\sum_{k=1}^{32} S_{i,k}/T_k\right)/32$. The standard deviation of the rates (indicated by error bars) over these 32 trials are computed by $\left(\sum_{k=1}^{32} \left(S_{i,k}/T_k\right)^2 - \bar{\beta}_i^2\right)/(32 - 1)$. Parameters: $a = 0.1$, $h = 5$ Hz, $\tau = 10$ ms.

timescales, the dynamics is dominated by the relaxation time to the pseudo-equilibrium it occupies. At large timescales, the dynamics is dominated by sharp transitions between distinct pseudo-equilibria. In the TMF approach, the rate of transitions between pseudo-equilibria vanishes exponentially fast with the size of the system, and the system becomes multistable in the infinite-size limit [35]. We expect a similar picture in the RMF approach. To validate this expectation, we consider the RMF limit for the simplest metastable dynamics, that of a neural-network model alternating between only two pseudo-equilibria. We expect to detect multistability via a transition whereby the RMF system Eq (20) starts admitting several stable solutions. In practice, the stability of a solution can be checked by creating an ensemble of iterations starting from randomized initial values.

**Multistable limit of metastable networks.** In principle, elementary computations can unfold in neural circuits by allowing some inputs to gate transitions between distinct output states of the circuit. In noisy neural networks, such gating processes give rise to metastable dynamics [18, 73–75]. Metastability has been studied for neural networks in the TMF limits [35, 76]. Here, we extend the analysis of metastability to the RMF framework with of focus on neural variability. Specifically, we demonstrate that our computational approach can capture the bistable response of a network model used to emulate perceptual rivalry [36].

The structure of the network is shown schematically in Fig 10a. The network comprises two symmetric groups of neurons $Group_1$ and $Group_2$, with identical features and symmetric connections. Each group comprises a cluster of excitatory neurons ($Exc_{1,2}$) and a cluster of inhibitory neurons ($Inh_{1,2}$). The excitatory neurons in $Exc_{1,2}$ are fully connected within their own clusters, represented by the self-pointing arrows. The inter-cluster arrows represent full connection between corresponding clusters. For simplicity, we set all the excitatory and inhibitory synaptic strengths to be equal to $\mu_e$ and $\mu_i$, respectively. Aside from their within-network interactions, all neurons are subjected to TMF-type inputs of fixed rates and strength with $(\beta\mu)_{TMF} = 1.5$ kHz. This corresponds, e.g., to each neuron having 1000 weak synapses of strength $\mu_{TMF} = 0.1$ activating at a rate of $\beta_{TMF} = 15$ Hz. Accounting for these TMF-type inputs amounts to adding a linear term in the function $V(u) \rightarrow V(u) + (\beta\mu)_{TMF} u$. This hybrid picture

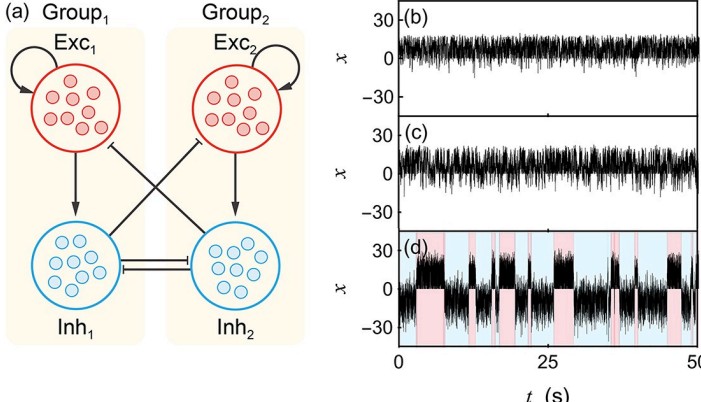

**Fig 10. Network structure and simulated time series of internal variable $x$.** Panel (a) shows that the system has two identical groups ($Group_{1,2}$) of neurons with symmetric connections. In each group, there is one cluster of excitatory neurons and one cluster of inhibitory neurons. Each cluster consists of an equal number of $K/4$ neurons. The connections between clusters are as shown. The synaptic strength is $\mu_e$ from any excitatory neuron and $\mu_i$ from any inhibitory one. In panel (b), (c) and (d), we plot the time series of $x$ for a representative neuron in $Exc_1$, with fixed inhibitory strength $\mu_i = -4.0$. The system exhibits: in panel (b), monostability with $\mu_e = 0.7$; in panel (c), fluctuation with $\mu_e = 1.2$; in panel (d), bistability with $\mu_e = 1.7$. Parameters: $K = 40$, $h = 1$ Hz, $a = \ln(100)/20 \approx 0.23$, $\tau = 10$ ms.

is biologically relevant considering recent evidence that only a restricted set of inputs is responsible for neuronal tuning, with most synapses only engaging in background activity [9, 37]. The inclusion of strong mutual inhibition across symmetric groups of neuron shall turn the network into a stochastic bistable switch. In the following, we study such a switch for a small total number of neurons ($N = 40$ with 10 neurons in each cluster), so that neural variability will be a key determinant of the overall dynamics. The ability to deal with such small systems is one of the core benefits of the RMF framework.

We first confirm through simulations that this system can indeed exhibit bistability. Fig 10b, 10c and 10d show the time series of the internal variable $x$ of one representative neuron in $Exc_1$. In Fig 10b, we simulate with $\mu_e = 0.7$ and $\mu_i = -4.0$. In this case, all activities are strongly suppressed by the inhibition. The system exhibits no bistability. Upon increasing the excitatory strength to $\mu_e = 1.2$ while leaving $\mu_i$ unchanged, the system becomes marginally bistable as shown by the enhanced fluctuations observed in Fig 10c. This is an indication that the corresponding deterministic system is on the edge of a dynamical transition or bifurcation. However, such bifurcations are notoriously difficult to detect in a noisy setting [38]. Further increasing $\mu_e$ reveals clear alternations between two relatively stable states as shown in Fig 10d for $\mu_e = 1.7$. We refer to these states as the "up" state or the "down" state depending on the mean value of $x$ during these states. Fitting the simulated dynamics via a hidden Markov model allows us to parse out the various dominance periods during which either $Group_1$ or $Group_2$ is up [77, 78]. This approach leverages the fact that conditionally to be up or down, the distribution of the internal variable $x$ remains approximately Gaussian.

Next, we check that our RMF framework can capture the bistable switch behavior, and perhaps even detect the bifurcation. To this end, we numerically solve the RMF system Eq (20) obtained for the considered network. As expected, we find two distinct stable solutions for high enough cross-inhibition $\mu_i$. When the RMF limit is multistable, different choices of initial values can lead to distinct solutions. Not surprisingly, if we choose larger initial values for neurons in $Group_1$ (e.g., $\beta_{0,i} = h + \epsilon, i \in Group_1, \epsilon > 0$), the iteration will converge to a state where $Group_1$ is in the up state. The thus obtained two sets of rate solutions parametrize two probabilistic models for two up and down metastable states.

In Fig 11, we compare exact simulation results in the finite-size network (data points) to our theoretical calculations in the infinite-size RMF limit (solid curves). The branching behavior of $\beta = \mathbb{E}[\lambda]$, $\mathbb{E}[x]$ and $\mathbb{E}[x^2]$ clearly indicates a transition from a monostable regime to a bistable regime. Our calculation is in good agreement with the simulated values outside of the shaded region. In the shaded region, where the bifurcation between monostability and bistability occurs, it is numerically ill-posed to distinguish between up and down states. The system does not persist in either states long enough for us to accurately compute the desired quantities. Fortunately, and as in the TMF limit, our theoretical calculation offers to precisely pinpoint the bifurcation via the forking behavior of the solution rates. However, contrary to the TMF limit, the RMF limit allows us to retain the neural variability due to stochastic inputs. In particular, at the cost of neglecting correlations, the RMF approach provides us with estimates about the higher moments of the dynamics. In Fig 11c, we show that the simulated and theoretically calculated second moments $\mathbb{E}[x^2]$ are well matched.

The absence of correlation in our RMF model appears not to affect our prediction, whereas Fig 12a shows that the original network exhibits a non-trivial correlation structure. This is because the non-trivial correlation structure of the original network is largely due to stochastic switching and simplifies to a tractable feedforward correlation structure when conditioning on the group of neurons that is up or down, as shown in Fig 12b and 12c. During a dominance period of the original dynamics, up-state neurons constitute the main source of variability. Fig 12b and 12c show that the activity of up-state neurons is essentially uncorrelated, in part

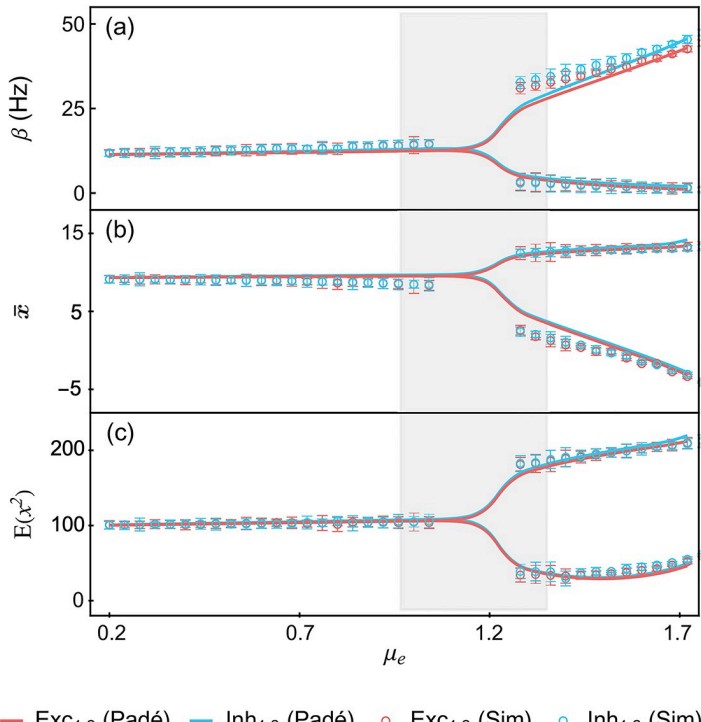

**Fig 11. Mean firing rates $\beta$, mean internal variables $\bar{x}$ and the second moments of the internal variables $\mathbb{E}[x^2]$ for the bistable networks from RMF calculations and simulations.** These figures shows in panel (a), the output rates $\beta$; in panel (b), the mean internal variables $\bar{x}$ and in panel (c), the second moments of $x$, with varying excitation strength $\mu_e$ and fixed inhibitory strength $\mu_i = -4.0$. Gray-shaded region is where the transition from monostable system to bistable system happened and we are not able to compute the simulated rates accurately, however, we can still perform RMF calculation within this region. For the simulated data, we run the event-driven simulation until 16000 spiking events for the entire network are accumulated. This process is repeated 32 times. During each repetition, we record the total time $T_k(k = 1, \cdots, 32)$ and total spiking event count $S_{i,k}$ for each neuron cluster $i(i = 1, \cdots, 4)$. The mean firing rates of neurons are computed by $\bar{\beta}_i = (\sum_{k=1}^{32} S_{i,k}/T_k)/32$. The standard deviation of the rates (indicated by error bars) over these 32 trials are computed by $(\sum_{k=1}^{32} (S_{i,k}/T_k)^2 - \bar{\beta}_i^2)/(32 - 1)$. Meanwhile, we record the entire time series of the internal variable $x$, with which we can compute the moment of $x$ for any given order. The standard deviations of the quantities (indicated by error bars) are computed over these 32 repetitions. Parameters: $K_{\text{total}} = 40$, $h = 1$ Hz, $a = \ln(100)/20 \approx 0.23$, $\tau = 10$ ms.

owing to their frequent but irregular spiking resets. Such an activity is well-approximated in the RMF limit. By contrast, the activity of down-state neurons is dominated by strong, shared feedforward inhibitory inputs. Fig 12b and 12c show that as a result, the activity of down-state is strongly correlated but in response to an approximatively Poissonian drive. Such an activity is also well-approximated in the RMF limit. More generically, we expect the RMF approximation to perform well for metastable systems with more than two quasi-equilibria, as long as a distinct quasi-equilibrium corresponds to a distinct population of neurons silencing all other populations of neurons.

**Finite-size effects and metastability.** Synaptic distributions in real neural networks follow a highly skewed distribution: there is a large number of weak connections but only a few strong ones, whose strengths vary over two orders of magnitudes [11]. Studies have also shown that meaningful neural activity is triggered by strong, correlated synaptic inputs rather than uncorrelated, weak ones [9, 37]. Our RMF modeling approach allows us to quantify how the size of the synaptic inputs impacts bistability, at the cost of neglecting activity correlations.

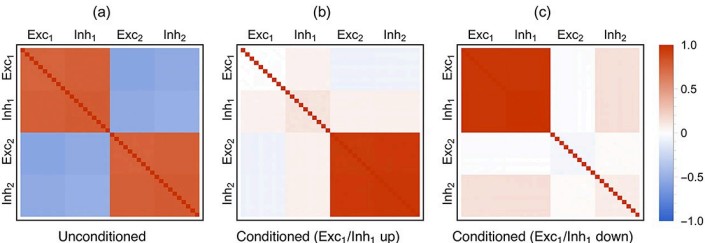

**Fig 12. Correlation structure of the original bistable network.** Panel (a) reveals that neurons are positively correlated within the same group, but negatively correlated across groups. This shows that the correlation structure of the metastable dynamics is primarily shaped by stochastic switches between up and down states. Panels (b) and (c) show the correlation structure of the same dynamics but conditioned on (b) $Group_1$ being up or (c) $Group_1$ being down. This shows that up-state neurons are weakly correlated, whereas down-state neurons are strongly correlated for receiving strong, shared inhibitory inputs. The correlations are estimated from simulated data, where we run the event-driven simulation until 400000 spiking events for the entire network are accumulated. We record the time series of the internal variables $x_i$. The unconditioned correlations are computed by $Cor(x_i, x_j) = Cov(x_i, x_j)/(Std(x_i)Std(x_j))$. The conditioned correlations are computed using the corresponding time series components (up/down state) detected and cropped from the whole time series. Parameters: $\mu_e = 1.5$, $\mu_i = 4.0$, $a = \ln(100)/20 \approx 0.23$, $h = 1$ Hz, $\tau = 10$ ms.

To do so, we consider the same hybrid model where neurons are all subjected to TMF-type background inputs, but where the circuit connections implementing cross-inhibition are treated in the RMF limits.

In this setting, we can quantify the emergence of bistability when varying the synaptic strength $\mu_e$ and $\mu_i$ involved in the cross-inhibitory circuit, thereby performing a phase space analysis of the network. Importantly, this phase space analysis only requires solving the corresponding self-consistent RMF equations rather than prohibitively costly simulations. We utilize the RMF solution rates to quantify bistability via the following bifurcation observable $\Delta$:

$$\Delta = \frac{\beta_{Exc_u} + \beta_{Inh_u} - \beta_{Exc_d} - \beta_{Inh_d}}{\beta_{Exc_u} + \beta_{Inh_u} + \beta_{Exc_d} + \beta_{Inh_d}}. \tag{24}$$

where the subscripts $Exc_u$, $Inh_u$ and $Exc_d$, $Inh_d$ refer to neuronal groups in the up and down states, respectively. By definition, as both excitation and inhibition are larger in the up state, the observable $\Delta$ ranges between 0 and 1. In the monostable regime, $\Delta \approx 0$ is minimum since $(\beta_{Exc_u}, \beta_{Inh_u}) \approx (\beta_{Exc_d}, \beta_{Inh_d})$. By contrast, in the bistable regime, when the neurons are silenced in the down state, i.e. $(\beta_{Exc_d}, \beta_{Inh_d}) \approx (0, 0)$, $\Delta \approx 1$ is maximum. Fig 13a shows the parametric density plot of $\Delta$ obtained by varying $\mu_e$ and $\mu_i$.

As expected, the regime of activity of the system is controlled by the overall level of cross-inhibition, which appears loosely linear in $\mu_e$ and $\mu_i$. This is because cross-inhibition obviously requires the excitation of the mediating inhibitory neurons. For weak cross-inhibition, the network response is dominated by the uniform TMF background and fluctuates around its only equilibrium state. Increasing $\mu_e$ and $\mu_i$ leads to a sharp transition to a bistable regime. For small circuits, with about 10 neurons per subnetworks, this transition occurs for large synaptic weights requiring an RMF treatment as the TMF approximation yields wrong rate estimates. The sharpness of the transition indicates a strong silencing of neurons in the down state, in line with the strong nonlinearity of the network dynamics. In that respect, we find that the TMF approximation predicts that bistability emerges for smaller synaptic weights than observed in small networks, while also underestimating the firing rates.

We conclude by utilizing our RMF computational approach to exhibit behaviors that are otherwise challenging to obtain via simulations. Specifically, we exhibit in Fig 13b the scaling

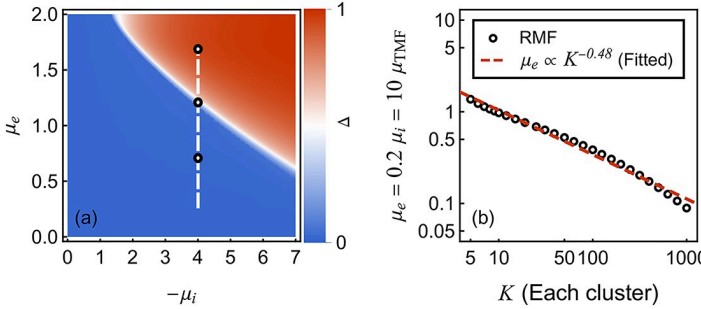

**Fig 13. Phase transition.** Panel (a): In this density plot of bifurcation observable $\Delta$, $\mu_e$ and $\mu_i$ are varied. The white dashed line represents the parameters $(\mu_e, -\mu_i)$ used in Fig 11. The black circles correspond to the parameters for the simulations shown in Fig 10b, 10c and 10d. Panel (b) is the log-log plot of the value of $\mu_e$ at the onset of bifurcation varying the size of the network, determined by thresholding $\Delta = 0.01$ in the RMF calculation (black circles). The red dashed line is the linear fitting in the log-log scale, which has a slope of $-0.48$. Parameters: $h = 1$ Hz, $a = \ln(100)/20 \approx 0.23$, $\tau = 10$ ms.

of the critical weights at the bifurcation $\mu_e$ with respect to the subnetwork size $K$, assuming a fixed ratio $\mu_e/\mu_i = 0.2$ and $\mu_e/\mu_{\mathrm{TMF}} = 10$. Interestingly, we found that the critical weight scales approximately as $1/\sqrt{K}$, similar to the scaling formally defining the balanced thermodynamic limit. In this limit, inhibition and excitation cancel one another on average. As a result, neuronal activity is driven by the fluctuations in the inputs. The $1/\sqrt{K}$ scaling ensures that neural variability is preserved in the large $K$ limits for independent inputs. We observe a similar phenomenon for critically tuned networks in the RMF limit, for which inputs are always independent. This is because neural variability is an invariant of the dynamics at the bifurcation, where networks start alternating between the newly individualized up and down states. Preserving this variability throughout network sizes in the RMF limit naturally requires a $1/\sqrt{K}$ scaling, which remarkably persists down to very small system sizes.

## Discussion

### Modeling assumptions for a computational framework

One of our leading motivations is the hope to quantify finite-size effects in the dynamics of noisy neural networks. To this end, we have adopted a reductionist modeling approach building on the central assumption that noise arises internally via a rate-based, spike-generating process. To account for the potential individual impact of a single synaptic activation, we have modeled the instantaneous spiking rate as a history-dependent stochastic intensity $\lambda$. Each synaptic delivery transiently impacts this stochastic intensity by causing instantaneous jumps in keeping with the synaptic weights. This bare framework leads to a series of well-identified pitfalls, which can be fixed with additional modeling components.

First, the requirement that the stochastic intensity remains a nonnegative quantity imposes that the joint integration of inhibition and excitation should be mediated by an internal variable $x$. This variable, albeit thought of as a membrane potential, is allowed to vary without bound, so that inhibition and excitation can be safely integrated algebraically. In turn, the stochastic intensity is deduced from $x$ via a necessarily nonlinear, rectifying function, akin to the $f - I$ curves. The simplest such function that is also monotonic is the exponential one [31, 79].

Second, with supralinear rectifying functions, the dynamics of recurrent networks can become ill-posed, with possibly diverging stochastic intensities [80]. Such an explosive behavior can be tamed by introducing an additional nonlinearity in the rectifying functions, e.g., by

imposing biologically plausible high-rate saturation. An alternative route that does not require saturation is to consider an instantaneous post-spiking reset mechanism, which implements a form of refractory period [62, 63]. By contrast with that implied by saturation, the type of non-linearity introduced by this reset mechanism is still computationally tractable.

Third, in the absence of relaxation, neuronal dynamics may be ill-posed when neurons are dominated by inhibitory inputs, even in the presence of a reset mechanism. Indeed, steady inhibition can cause the internal variable to diverge over time $\lim_{t \to \infty} x(t) = -\infty$, effectively silencing the neuron. We remedy such a caveat by allowing the internal variable to relax toward a base level. While biophysically relevant, such relaxation precludes runaway dynamics, and thereby neuronal silencing. Incidentally, it also guarantees the ergodicity of the finite-size dynamics [23].

## Biophysical relevance and limitations

Theoretical considerations alone show that all the above modeling assumptions must be included in any stochastic-intensity-based neural models. These are also the only modeling features we consider. This drastic oversimplification clearly neglects important aspects of neuronal processing such as propagation delays [81], synaptic adaptation and fatigue [82], or neuronal compartmentalization [83]. Perhaps the most serious limitation of the LER model is being current-based for integrating inputs without conductance mediation. In conductance-based models, the internal variable $x$ explicitly models the membrane voltage, which is naturally bounded by the ionic reversal potentials. In the context of point-process-based models, this naturally excludes the possibility of rate explosion. We do not address these limitations here as our main point is only to develop a computational framework where the impact of finite-size effects can be quantified in relation to a few key modeling assumptions.

That being said, despite the crudeness of their modeling assumptions, we found that LER neurons operate in a biologically relevant regime for parameters inferred from real-world measurements. In particular, we set large synaptic weights to be such that a single synaptic excitation at base level causes the internal variable $x$ to transiently increase by 2% of its range. Here we defined the range as the mean value of $x$ when $\beta \simeq 100$ Hz. Such choices mirror the observation that large synaptic events cause up to $\sim 0.5$ mV depolarization at the soma, for a typical upward voltage range of 20 mV. There are also natural choices for the two timescales featuring as free parameters [79]: We set the relaxation timescale $\tau$ to be equal to the membrane time constant 10ms. We adopted a base rate of $h = 1$ Hz, as the putative spontaneous firing rate of an isolated neuron. The latter choice of a base rate is actually more flexible than it appears. Indeed, varying levels of background activity modulates the effective value of $h$.

This biologically relevant computational framework opens up the possibility for the studies of novel biological phenomena. For example, we can estimate the parametric dependences of the experimentally measurable quantities (e.g., means and variances of the membrane voltages) on the finite-size network components (e.g., network sizes and synaptic strengths). This allows us to account for the role that individual neurons can play in shaping the neural activities, in contrast with most computational frameworks that ignore such individual impacts. Such characterization of the activities by the finite-size components further allows us to infer the underlying network structures, of which our computational framework can readily study the collective neural dynamics such as metastability.

Moreover, there are direct extensions to LER models that can be treated within our framework. One alteration is to implement a post-spiking hyperpolarization of fixed size, say $v$, rather than a hard reset to zero. In this case, the discrete derivative term shall be replaced by the delay term $L(a + u)(e^{uv} - 1)/u$, and the resulting DDEs are still tractable via resolvent

formalism. However, the rate-transfer function would become asymptotically linear so that network dynamics will no longer be unconditionally stable. Other alterations are to allow for relaxation toward a value $b \neq 0$ and resets to value $r \neq 0$. This leads to considering auxiliary terms of the form $V(u) = bu + \sum_j \beta_j (e^{u\mu_j} - 1)$ with spontaneous rate of the form $he^{ar}$. Again, the resulting RMF problems are still tractable but now offer some interesting modeling aspects. One such aspect is to allow for $b$ and $r$ to be slowly dependent on neuronal activity to model some form of a fatigue or adaptation process.

## Analyticity, nonlocality, and singular perturbation

The computational framework of the RMF approach relies on the Poisson hypothesis, which posits that neural inputs are distributed according to independent Poisson processes [28, 48]. Such a hypothesis is justified when neuronal interactions are randomized across an infinite number of replicas of a given finite-size network. The simplifying Poisson hypothesis allows one to parametrize the probability distribution of a neural network via the individual neuronal spiking rates alone. Therefore, in the RMF limit, elucidating the typical state of a neural network amounts to solving self-consistently for these stationary firing rates. These rates are determined as fixed-point solutions of the system of equations specified by rate-transfer relations, which can be seen as implementing some nonconservative Kirchhoff's laws [84].

To compute rate-transfer functions in the RMF limit, our strategy is to derive conservation laws holding in the stationary regime. Then, we use these conservation laws to functionally characterize the typical distribution of neuronal states via their MGFs. For LER neurons, such a functional characterization takes the form of nonlocal DDEs bearing on the MGFs $L$. Analytical solutions to these DDEs uniquely specify these MGFs, from which the stationary rates can be deduced as $\beta = hL(a)$. However, as for most nonlocal equations, solutions to our DDEs cannot be expressed explicitly and one has to resort to singular perturbative methods.

In principle, one can hope to obtain tractable distinguished limits in two asymptotic regimes: for vanishing relaxation $\tau \to \infty$ and for vanishing spontaneous rate $h \to 0$. In practice, only the limit $h \to 0$ is useful. Indeed, when $\tau \to \infty$, the DDE simplifies to a pure delay equation, which constitutes an ill-conditioned numerical problem. More fundamentally, in the absence of relaxation, we do not know whether the original dynamics is ergodic. In particular, the RMF limit may become ill-posed with neurons being silenced for diverging durations by increasingly large amounts of inhibition. By contrast, when $h \to 0$, the DDE becomes a simple ODE, so that the corresponding MGF $L_{h=0}$ is given as the only solution satisfying the normalization condition $L_{h=0}(u) = 1$. This solution is valid whenever the reset mechanism can be neglected, i.e., for low output spiking rate $\beta \ll 1/\tau$.

For moderately large firing rates, we are able to solve the DDEs via resolvent formalism. Interestingly, this resolution does away with the requirement of specifying initial conditions on a delayed interval, as one generally expects for DDEs. This is because the resolvent formalism naturally selects for the set of solutions such that the graph of $(h, u) \mapsto (h, L(h, u))$ forms a continuous manifold anchored on the analytical boundary $L_{h=0}$ when $h \to 0$. Thus, we circumvent the need for local initial conditions over an interval for $L_h$ at fixed $h$, by imposing global regular conditions on $(h, u) \mapsto (h, L(h, u))$ when varying $h$. We conjecture that this manifold is analytic in $u$ as required for MGF functions, whereas analyticity in $h$ can fail, at least in $h = 0$ when excitation dominates the input drives. Because of the lack of analyticity in $h$, the resolvent formalism only produces a formal, possibly divergent, series expansion in terms of powers of $h$, the singular perturbation parameter. Although we have not characterized the type of non-analyticity at stake, we found that Padé approximants summation can accurately predict output rates when the formal series diverges. This suggests that $L(\cdot, u)$ might be a meromorphic

function on some open region of the complex plane whose singularities all lie outside of the nonnegative real axis [85].

## Dynamical transition via RMF limits

We utilize our RMF framework to partially analyze the metastable dynamics of small networks. As well-known for the TMF limit, metastability turns into multistability in the RMF limit: the residence times in the various pseudo-equilibria diverge exponentially with the number of replicas. By contrast with the TMF limit, however, these pseudo-equilibria remain probabilistic, with stationary distributions parametrized by self-consistent firing rates. Moreover, the RMF approach predicts the transition to bistability in small metastable networks with accurate estimates of the mean activity as well as the neural variability, when conditioned to being in either pseudo-equilibrium.

Detecting bifurcation in stochastic systems is a notoriously arduous task [38]. This is because including noise in a finite-dimensional system generally turns its dynamics into an infinite-dimensional one, typically specified via a master equation. For instance, one can show that the transient dynamics of a $K$-neuron LER network is specified by a linear partial differential equation bearing on the time-dependent MGF $L(t, \boldsymbol{u})$:

$$\partial_t L + \mathcal{G}[L] = 0, \quad \text{with} \quad L(t, \boldsymbol{u}) = \mathbb{E}\left[e^{\sum_{i=1}^{K} u_i x_i(t)}\right], \tag{25}$$

where $\mathcal{G}$ is some $K$-dimensional nonlocal differential operator. Loosely speaking, ergodic metastable networks can be characterized as those networks admitting multimodal stationary distributions, which in principle, can be derived from the knowledge of $L(t, \boldsymbol{u})$. Unfortunately, due to its nonlocalities, Eq (25) as well as its stationary version $\mathcal{G}[L] = 0$, is impervious to an analytical and numerical treatment. These hindering nonlocalities are due in part to the possible nonlinearity of the stochastic intensities, but mainly to the boundary terms mediating neuronal interactions [23, 52].

The RMF strategy consists in considering approximate infinite-size networks where these boundary terms simplify thanks to the Poisson hypothesis. According to the Poisson hypothesis, neurons behave independently so that the transient MGF admits a product form $L(t, \boldsymbol{u}) = \prod_{i=1}^{K} L_i(t, u)$, where each marginal MGF $L_i(t, u) = \mathbb{E}[e^{u x_i(t)}]$ satisfies a nonlocal partial differential equation

$$\partial_t L_i(t, u) + \frac{\partial_u L_i(t, u)}{\tau_i} - \frac{V_i(t, u)}{u} L_i(t, u_i)$$
$$+ h_i\left(\frac{L_i(t, u + a_i) - L_i(t, a_i)}{u}\right) = 0. \tag{26}$$

The resolution of the above equation still resists direct treatment in the transient regime. However, in the stationary regime, the infinite-dimensional PDE problem reduces to a DDE problem, which is revealed to be finite-dimensional via our analysis: all boils down to solving the $K$-dimensional RMF system (10). This finite-dimensional setting allows to resort to classical bifurcation analysis [86]. Indeed, solving Eq (10) shows that the transition to bistability occurs in the RMF limit when the single equilibrium obtained for low cross-inhibition loses stability and when two distinct pseudo-equilibria emerge. Thus, the emergence of bistability is dynamically akin to a classical, finite-dimensional supercritical pitchfork bifurcation in the RMF limit.

However, it is perhaps better stated to say that the RMF framework detects metastability as a static phase transition rather than a dynamical bifurcation. After all, our RMF treatment only considers stationary solutions with no explicit regard for the transient dynamics, while the

emergence of bistability corresponds to the spontaneous ordering of all replicas. In this light, we predict that the emergence of bistability in the RMF limit represents a continuous phase transition. This prediction is supported by the apparent continuity of the stationary rates through criticality, which is to be expected from the bifurcation picture. Moreover, the bifurcation picture also suggests that the RMF phase transition is analogous to low-temperature spontaneous magnetization in the Curie-Weiss model, with cross-inhibition weights playing the role of coupling variable. However, the order of the phase transition remains to be elucidated as numerics suggest a rather smooth dependence of the rates at criticality. It is also worth noting that critical RMF networks could exhibit nontrivial long range correlations across replicas, reminiscent of second-order phase transition. It remains to be explored whether such critical behavior would encode any information about the original metastable dynamics.

More generally, we hope to leverage our RMF framework to estimate key dynamical properties of metastable dynamics, such as the transition rates between pseudo-equilibria. In the classic theory of reaction rates, these transition rates can be approximated via phenomenological Eyring-Kramer's laws given some notion of energy landscapes [87]. Unfortunately, we do not have a consistent notion of energy in our model. Nevertheless, we can still use the first and second moments of the internal variable $x$ to form Eyring-Kramer's-type candidate laws for the observed rates. Although we do not have a systematic derivation for such laws, preliminary analysis suggests that these transition rates can be inferred from our RMF predictions. In support of this, Fig 14 shows that the empirical transition rates observed in our bistable system are accurately predicted from our RMF treatment. The empirical rates are estimated from exact but numerically expensive Monte-Carlo simulations, whereas the RMF rates are computed

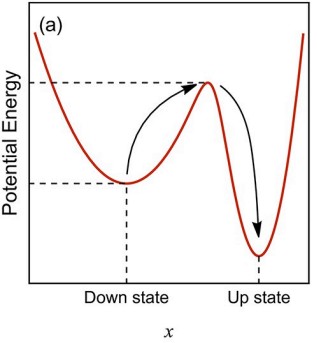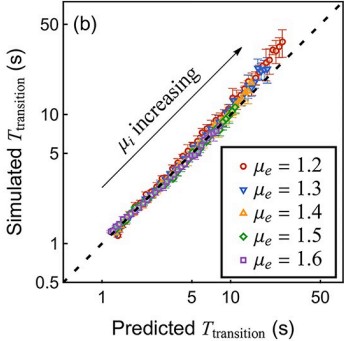

**Fig 14. Prediction of transition times using Eyring-Kramer's law.** Panel (a) represents schematically the effective double-well potential energy landscape used to predict the transition rate for the bistable network. In principle, such an energy landscape should at least be four-dimensional as it involves 4 groups of neurons. However, an one-dimensional effective model already offers good predictions as we find that a single switching direction (arrow) dominates the transition dynamics. Panel (b) shows that the transition times inferred from our RMF model (horizontal axis) predict the empirical transition times obtained from Monte-Carlo simulations (vertical axis). Each data point corresponds to a different pair of parameters $(\mu_e, \mu_i)$. $\mu_e$ ranges from 1.2 to 1.6 in the increment of 0.1. Data points with distinct $\mu_e$'s are shown in different colors and marks. $\mu_i$ ranges from 4 to 14 in the increment of 0.25. The arrow in the figure indicates the direction of increasing $\mu_i$. For the predicted transition times, we apply the Eyring-Kramer's law by estimating the potential energy of our system from a mixture of two normal distributions. Such distribution is constructed using solely the means and standard deviations of the up and down states computed from the RMF calculation. Please refer to S1 Appendix for more details. For the simulated transition times, we simulate the dynamics of our bistable network until 100 transitions between up and down states are accumulated. The transitions are detected using a hidden Markov model. This process is repeated 16 times. During each repetition, we record the total time $T_k (k = 1, \cdots, 16)$. The simulated transition times of the network are then computed by $T_{\text{transition}} = (\sum_{k=1}^{16} T_k / 100)/16$. The standard deviations of the transition times (indicated by error bars) over these 16 repetitions are computed by taking the square root of $(\sum_{k=1}^{16} (T_k/100)^2 - T_{\text{transition}}^2)/(16 - 1)$. Parameters: $K_{\text{total}} = 40$, $h = 1$ Hz, $a = \ln(100)/20 \approx 0.23$, $\tau = 10$ ms.

phenomenologically via an Eyring-Kramer's-type law. This law is obtained from a simple one-dimensional energy landscape model inferred from the knowledge of the first and second moments of the internal variable $x$ conditioned to being in the up or down state (see S1 Appendix for details). To justify the form of our one-dimensional model, we heuristically determine that switching between up and down states is dominated by a single escape process, whereby down-state neurons transiently activate by chance. The preliminary results presented in Fig 14 demonstrate the possibility of predicting the transition rates with our RMF framework. Further quantifying the dependencies of the transition rates on all the modeling parameters will require a more principled treatment. We anticipate that such treatment will extrapolate finite-size transition rates from the scaling behavior of the corresponding rates in the RMF limit, which shall satisfy some large deviations principle [88].

## Supporting information

**S1 Appendix. Supplementary text.** A: Derivation of the DDEs, B: Simulation method, C: Calculation for higher moments, D: Different approximation methods, E: Prediction of dynamical transition rates. Fig A: Bistable networks. Fig B: State switching paths.
(PDF)

**S1 Fig. Distribution of the internal variable $x$.** The distribution of the internal variable $x$ of a single neuron subjected to various different types of inputs. The detailed input parameters are listed above. Parameters: $h = 1$ Hz, $a = 0.1$, $\tau = 10$ ms.
(PDF)

## Acknowledgments

We would like to thank François Baccelli, Michel Davydov, Yiran Hu, Zhao Liu, Manyi Yim for insightful discussions.

## Author Contributions

**Conceptualization:** Luyan Yu, Thibaud O. Taillefumier.

**Data curation:** Luyan Yu.

**Formal analysis:** Luyan Yu, Thibaud O. Taillefumier.

**Funding acquisition:** Luyan Yu, Thibaud O. Taillefumier.

**Investigation:** Luyan Yu, Thibaud O. Taillefumier.

**Methodology:** Luyan Yu, Thibaud O. Taillefumier.

**Project administration:** Thibaud O. Taillefumier.

**Software:** Luyan Yu.

**Supervision:** Thibaud O. Taillefumier.

**Validation:** Luyan Yu, Thibaud O. Taillefumier.

**Visualization:** Luyan Yu, Thibaud O. Taillefumier.

**Writing – original draft:** Luyan Yu, Thibaud O. Taillefumier.

**Writing – review & editing:** Luyan Yu, Thibaud O. Taillefumier.

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
