## [Decision Letter · Decision Letter 0]

11 Dec 2021

Dear Dr. Taillefumier,

Thank you very much for submitting your manuscript "Metastable spiking networks in the replica-mean-field limit" for consideration at PLOS Computational Biology.

As with all papers reviewed by the journal, your manuscript was reviewed by members of the editorial board and by several independent reviewers. In light of the reviews (below this email), we would like to invite the resubmission of a significantly-revised version that takes into account the reviewers' comments.

Specifically, both the editor and reviewers agreed that this work, while interesting and potentially of great utility, seems to be 'somewhat incomplete' in that the relevance to biological systems is hinted at but not demonstrated.  I would encourage the authors to enhance their manuscript to make it both more accessible and more clearly relevant to the neuroscientific community at large.  Reviewer 1, in particular, laid out an excellent strategy for revision that I believe would greatly increase the impact of this work.  Also of use would be an an example of a more sophisticated application of their approach to a problem of interest to computational neuroscientists like memory retrieval or decision making via evidence integration that leads to a phase transition to an attracting state representing the agents action.  

We cannot make any decision about publication until we have seen the revised manuscript and your response to the reviewers' comments. Your revised manuscript is also likely to be sent to reviewers for further evaluation.

Sincerely,

Jeff Beck

Associate Editor

PLOS Computational Biology

Lyle Graham

Deputy Editor

PLOS Computational Biology

Reviewer's Responses to Questions

**Comments to the Authors:**

Reviewer #1: Summary:

In this manuscript, Yu and Taillefumier extend prior work by the second author and other colleagues on Replica Mean Field (RMF) limits of Galves-Locherbach intensity-based models of neural networks to models that can support metastability. They devise an approximation method to numerically interrogate the stationary statistics of networks in this limit, and show that this method yields reasonably accurate predictions when compared against direct Monte Carlo simulations. The motivating question---How do metastable states emerge in finite networks of spiking neurons, and what are their properties?---is of theoretical interest, and the math appears correct. However, the original contributions of the paper are not sufficiently clear, nor is the writing sufficiently accessible, to justify publication of its present form in PLoS Computational Biology.

Major comments:

1. The lack of novel biologically-relevant insights in the present manuscript renders it somewhat unsatisfying. While the technical component of this work presents a well-defined incremental advance relative to prior work by the second author (notably Refs. [25] and [46]), the main result is an approximate numerical framework which does not in of itself yield substantial insight. I am not convinced that the solution method---which relies upon a combination of relatively standard perturbative methods---is sufficiently innovative to justify publication without further evidence of its broad utility. The scientific contribution of the present work would be much clearer if the authors could show more clearly that their methods yield novel biologically-relevant insights beyond the possibility of detecting the transition to bistability.

One clear way for the authors to enhance the impact of this paper would be to at least partially realize their ``hope [of leveraging the] RMF framework to estimate key dynamical properties of metastable dynamics'' (lines 858-859). I found it somewhat frustrating that the authors reported a preliminary result that would clearly support the idea that their framework can provide useful insights towards the stated goal of understanding metastability (namely, that ``the static approach of the RMF framework is predictive of the transition rates,'' lines 866-867) in the Discussion. This leaves the reader with the impression that the present work is somewhat incomplete. I would therefore suggest that the authors take the time to treat this point carefully, and then submit a manuscript containing a more scientifically complete story. Alternatively, they might consider submitting the present manuscript to a more specialized journal.

2. Some revision to the prose would be required in order to make the paper accessible to PLoS Comput. Biol.'s broad audience. While this referee certainly appreciates the difficulties inherent in crafting a manuscript that is both technically rigorous and broadly accessible, the balance struck by the current version could be improved. Given the fact that the second author's prior publications on the RMF method provide the requisite technical details rigorously, one option would be for the authors to take this opportunity to give a more pedagogical introduction to these ideas before detailing the current technical advance. The authors might alternatively consider re-organizing the manuscript to emphasize biologically-relevant results before describing some of the methodological details in full.

Even for a reader relatively well-versed in the required mathematics and neuroscience, the readability of the manuscript is compromised by the distracting frequency of typos and grammatical errors.

3. As the RMF method results in self-consistent equations that must ultimately be approximated and solved numerically through a perturbative scheme, it is important that the authors provide sufficient justification for why this method is advantageous relative to direct Monte Carlo simulation. Some examples of where this point should be reified are as follows:

a. Lines 433-434: ``Bear in mind that in all cases, our computational results are obtained incomparably faster than those estimated via Monte-Carlo simulations." This claim---particularly given the hyperbolic descriptor ``incomparably''---should be supported with explicit runtime statistics, as well as a detailed description of the computational resources used to generate the presented results.

b. Lines 678-679: ``We conclude by utilizing our RMF computational approach to exhibit behaviors that are otherwise challenging to obtained via simulations.'' Please provide a more detailed argument to justify this claim.

As mentioned above, it is also important---particularly given the readership of PLoS Comput. Biol.---to argue that this method has strong potential to generate novel biological insights that could not easily be gleaned from Monte Carlo simulations.

Minor Comments:

1. Figure 1 appears to be identical to Figure 4 of Ref. [46] (Baccelli and Taillefumier, SIAM J. Applied Dynamical Systems 2021); reproduction should be noted.

2. Lines 277-278: Can you provide evidence for this conjecture?

3. Figures 4-10: Please specify sample sizes in each figure caption, and indicate precisely how error bars are computed. Some information is given in the caption for Figure 4, but it is not sufficient.

4. Lines 639-642: Please elaborate on this point.

5. Line 750: I cannot follow the usage of ``perspective'' in this context.

6. The legibility of the figures could be improved by making them colorblind-friendly and making the error bars more distinct.

Reviewer #2: In this well written piece, the authors pursue their line of research on the introduction of replica methods to describe the dynamics of finite (and potentially small) random neural networks. This research is certainly interesting and opens the way to new discoveries, some potentially interesting for biology. In this paper, the authors extend their methodology (both theoretically and numerically) to address the question of the characterization of metastable states in networks that include excitation, inhibition and exponential firing rates. The originality and main contribution of the paper lies in a clever use of the "resolvent formalism" instead of reduction to a delay equation that was used in previous works; this method for solving functional equations that involves infinite series that may be divergent, a problem clearly discussed in the manuscript. They combine this method to the use singular perturbation theory. This allows the authors to identify and describe pseudo-equilibria. Ample numerical simulations of the system are provided. They are performed with care, using advanced numerical methods based on Padé approximants that also appropriately handle possible divergences in Taylor series by considering rational fractions instead of polynomials, and that largely expand the range of convergence of numerical approximation.

The authors use these techniques to compute the nonlinear transfer function and their moments, with a particular focus on finite-size effects. They next look for fixed points of the equation and multi-stability, together with swtiches between equilibria in an E/I network, with a quite fine analysis of a (probably pitchfork or saddle-node) bifurcation of the system in response to changes in the excitation strength. The discussion is well thought and rich, and covers both questions related to the biological relevance of the findings, as well as further technical considerations that are deep and interesting. All derivations are provided in a supplementary file that is brief and clear. All derivations were checked and seem correct.

While the Referee likes this paper, it seems that this paper belongs more to the domain of stochastic analysis and theory rather than addresses a specifically formulated question in biology. The link with biology is well presented, but it did not appear to the Referee that this link was strong enough to warrant publication in Plos CB, whose mandate is to publish articles that "should model aspects of biological systems, demonstrate both methodological and scientific novelty, and provide profound new biological insights". Here I find that while methodological and scientific novelty, as well as rigor, are present, we are missing a clear and strong connection to biology both in the motivation and in the insight it provides.

**Have the authors made all data and (if applicable) computational code underlying the findings in their manuscript fully available?**

Reviewer #1: None

Reviewer #2: Yes

PLOS authors have the option to publish the peer review history of their article (what does this mean?). If published, this will include your full peer review and any attached files.

Reviewer #1: No

Reviewer #2: No
---

## [Decision Letter · Decision Letter 1]

21 Apr 2022

Dear Dr. Taillefumier,

Thank you very much for submitting your manuscript "Metastable spiking networks in the replica-mean-field limit" for consideration at PLOS Computational Biology. As with all papers reviewed by the journal, your manuscript was reviewed by members of the editorial board and by several independent reviewers. The reviewers appreciated the attention to an important topic. Based on the reviews, we are likely to accept this manuscript for publication, providing that you modify the manuscript according to the review recommendations.

JEFF:  You  really do need to fix the issue regarding Galves-Locherbach references mentioned in the reviewer comments below.  As the reviewer notes, in the computational neuroscience, this model is referred to as a linear non-linear Poisson (LNP) model with canonical reference to Pillow etal 2000 or a spike response model with canonical reference to Gerstner's book.  The simple fix is to use LNP throughout the manuscript and restrict a Galves-Locherbach reference to a single line in the introduction pointing out that some circles refer to it as Galves-Locherbach model.  

Regarding biological significance.  I very much appreciated the preview of your forthcoming work applying this technique and your reasons for withholding those results at this time.  However, I would urge you to expand your discussion by a single paragraph simply outlining 'how this work could be applied' without going into the details that you submitted in response to review.    

Sincerely,

Jeff Beck

Associate Editor

PLOS Computational Biology

Lyle Graham

Deputy Editor

PLOS Computational Biology

[LINK]

Reviewer's Responses to Questions

**Comments to the Authors:**

Reviewer #1: The authors addressed all my comments. I thank them for the thorough response. I recommend acceptance.

Reviewer #2: I appreciate the revisions of the paper and the efforts to enhance connections between the methodological contribution provided in this particular paper and concrete biological problems. The new sections added are also interesting. My main point related to this paper was the absence of a clear new biological insight. The authors have added some discussion about these aspects and also indicate future work in link with experiments. While this looks certainly interesting, credible and encouraging as of the applicability of this result to biology, I still believe that when considering the precise scientific contribution of this particular submission, the novelty and interest is more technical than biological and belongs to computer science or applied mathematics. To be more precise, even if possible connections are made more precise in the revised manuscript, I cannot assess that these revisions leads the paper to reach the scope of the Journal that "Research articles should [...] provide profound new biological insights.".

Moreover, I recommend the authors to consider in more detail the literature on neuron models. A quite surprising point of the paper that I had already noted in the first submission was the use of the terminology "Galves-Locherbach model" to refer to a classical Linear Nonlinear Poisson model, as introduced by Gerstner in 1994 and were popularized in the early 2000s by Pillow, Chichilnitsky and others. I checked the references cited and it indeed seem that Galves and Locherbach reinvented this model in a mathematical papers recently. In the frame of a submission to a Computational neuroscience journal (or in fact, any publication), I strongly suggest revising this terminology and that appropriate credit is given.

This being said, as a scholar in computational neuroscience with mathematical background, I appreciate this work and believe it is a well written paper that should deserve publication in a theoretical journal. I am also convinced that this methodology in nature will have the capacity down the line to yield interesting insight in neuroscience, but as of now, the contribution provided is technical in nature and warrants publication in a more specialized journal.

**Have the authors made all data and (if applicable) computational code underlying the findings in their manuscript fully available?**

Reviewer #1: None

Reviewer #2: Yes

PLOS authors have the option to publish the peer review history of their article (what does this mean?). If published, this will include your full peer review and any attached files.

Reviewer #1: No

Reviewer #2: No

Figure Files:

Data Requirements:

Reproducibility:

References:

---

## [Editor Report · Decision Letter 2]

16 May 2022

Dear Dr. Taillefumier,

We are pleased to inform you that your manuscript 'Metastable spiking networks in the replica-mean-field limit' has been provisionally accepted for publication in PLOS Computational Biology.

Best regards,

Jeff Beck

Associate Editor

PLOS Computational Biology

Lyle Graham

Deputy Editor

PLOS Computational Biology

---

## [Editor Report · Acceptance letter]

1 Jun 2022

PCOMPBIOL-D-21-01383R2 

Metastable spiking networks in the replica-mean-field limit

Dear Dr Taillefumier,

I am pleased to inform you that your manuscript has been formally accepted for publication in PLOS Computational Biology. Your manuscript is now with our production department and you will be notified of the publication date in due course.

With kind regards,

Zsofia Freund
